# Drones Help Drones: A Collaborative Framework for Multi-Drone Object Trajectory Prediction and Beyond

**Zhechao Wang**[1,2], **Peirui Cheng**[1], **Mingxin Chen**[1,2], **Pengju Tian**[1,2], **Zhirui Wang**[1*]
**Xinming Li**[1], **Xue Yang**[3], **Xian Sun**[1,2]
[1]Key Laboratory of Network Information System Technology,
Aerospace Information Research Institute, Chinese Academy of Sciences
[2]University of Chinese Academy of Sciences     [3]Shanghai AI Laboratory
{wangzhechao21, chenmingxin22, tianpengju22}@mails.ucas.ac.cn
yangxue@pjlab.org.cn
{chengpr, wangzr, lixm004499, sunxian}@aircas.ac.cn

## Abstract

Collaborative trajectory prediction can comprehensively forecast the future motion of objects through multi-view complementary information. However, it encounters two main challenges in multi-drone collaboration settings. The expansive aerial observations make it difficult to generate precise Bird's Eye View (BEV) representations. Besides, excessive interactions can not meet real-time prediction requirements within the constrained drone-based communication bandwidth. To address these problems, we propose a novel framework named "Drones Help Drones" (DHD). Firstly, we incorporate the ground priors provided by the drone's inclined observation to estimate the distance between objects and drones, leading to more precise BEV generation. Secondly, we design a selective mechanism based on the local feature discrepancy to prioritize the critical information contributing to prediction tasks during inter-drone interactions. Additionally, we create the first dataset for multi-drone collaborative prediction, named "Air-Co-Pred", and conduct quantitative and qualitative experiments to validate the effectiveness of our DHD framework. The results demonstrate that compared to state-of-the-art approaches, DHD reduces position deviation in BEV representations by over 20% and requires only a quarter of the transmission ratio for interactions while achieving comparable prediction performance. Moreover, DHD also shows promising generalization to the collaborative 3D object detection in CoPerception-UAVs. Our source code is available at https://github.com/WangzcBruce/DHD.

## 1   Introduction

Multi-drone object trajectory prediction [1–3] aims to collaboratively identify objects and forecast their future movements using multiple drones with overlapping observations, which can overcome the single-drone limitations of occlusions, blur, and long-range observations. Given the safety and reliability of task execution, the role of multi-drone object trajectory prediction is indispensable. It contributes to early warning of potential accidents and path planning in drone operations, better serving for intelligent cities [4], transportation [5], aerial surveillance [6] and response systems [7].

Current methods in collaborative object trajectory prediction are primarily categorized into two frameworks: multi-stage prediction [8, 9] and end-to-end prediction [10, 11]. The multi-stage paradigm achieves collaborative prediction based on individual perception results. Specifically,

---

*Corresponding author

it begins with object detection in each view, such as oriented object detection [12]. Then, the multi-view objects are correlated, and the generated trajectories are fed into a regression model for predictions. In contrast, end-to-end approaches transform each view's 2D features into BEV features at a unified 3D coordinate system and conduct feature-level correlation. Subsequently, future instance segmentation and motion are jointly predicted. The end-to-end paradigm significantly enhances accuracy and computational efficiency compared to multi-stage detect-track-predict pipelines, leading to its widespread popularity and application.

Nevertheless, end-to-end approaches are primarily designed for autonomous driving scenarios and may not directly apply to aerial perspectives. Specifically, they adopt the prevalent "Lift-Splat-Shoot" (LSS) [13] to predict pixel-wise categorical depth distribution and reconstruct objects' distribution in 3D space using the camera's intrinsics and extrinsics. In aerial contexts, observation distances are considerably more extensive, as depicted in Fig.1. This expanded range increases depth categories and poses substantial challenges to view transformation. Crucially, mistaken feature projection distorts BEV representations, severely undermining the reliability of subsequent multi-drone collaboration. Furthermore, efficient interaction is essential for real-time collaborative prediction. The prevalent sparse collaborative strategy, where2comm [14], utilizes the downstream detection head for information selection. However, its mode is inflexible and overlooks valuable contextual cues beyond the object information, which are equally essential for motion forecasting.

To address the challenge of view transformation, we observe that the drone's inclined observation leads to intersections between the sight and the ground plane. Consequently, this geometric attribute assigns a theoretical maximum depth to each pixel, providing a constraint for depth estimation. Moreover, given the noticeable gap between objects and the ground plane, it is an alternative to derive intricate depth estimation from the simpler height estimation. For the design of the collaborative strategy, we advocate that a flexible collaborative strategy should dynamically assess the information volume contributed by each region to downstream tasks based on the model's feedback. This adaptive methodology ensures a thorough understanding of the environment, encompassing both foreground objects and their broader surroundings, thus facilitating more accurate decision-making in collaborative trajectory prediction.

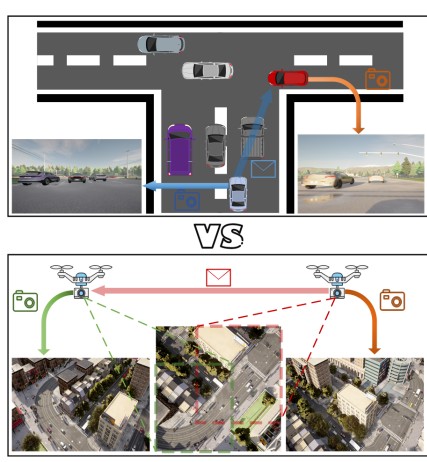

Figure 1: Comparative visualization of observations in autonomous driving versus aerial surveillance.

Building upon the mentioned solutions, this research presents a collaborative prediction framework for drones called DHD, consisting of a Ground-prior-based BEV Generation (GBG) module and a Sparse Interaction via Sliding Windows (SISW) module. The GBG module calculates the viewing angle of each pixel with the camera's intrinsic and extrinsic parameters. Subsequently, based on the flight altitude, this module determines the theoretical maximum depth for each pixel. With the guidance of the viewing angle and maximum depth information, a relatively simple height estimation can derive more precise depth, thus achieving more accurate BEV representations. The SISW module employs sliding windows to analyze discrepancies between the central and surrounding features, thereby quantifying the information volume at each position. Regions exhibiting significant feature variations are assigned higher information scores, indicating that they contain objects or crucial environmental information. Additionally, due to the lack of relevant datasets for multi-drone collaborative prediction, this study utilizes the CARLA [15] simulation platform to generate a novel dataset, "Air-Co-Pred", comprising cooperative observations from four drones across 200 varied scenes. This simulated dataset serves to validate the effectiveness of our proposed DHD framework in aerial collaborations.

In summary, our contributions are listed as follows:

A Ground-prior-based Bird's Eye View (BEV) Generation module is presented to achieve more accurate BEV representations guided by the ground prior and height estimation.

A Sparse Interaction via Sliding Windows module is proposed to improve the accuracy of multi-drone object trajectory prediction through efficient information interaction.

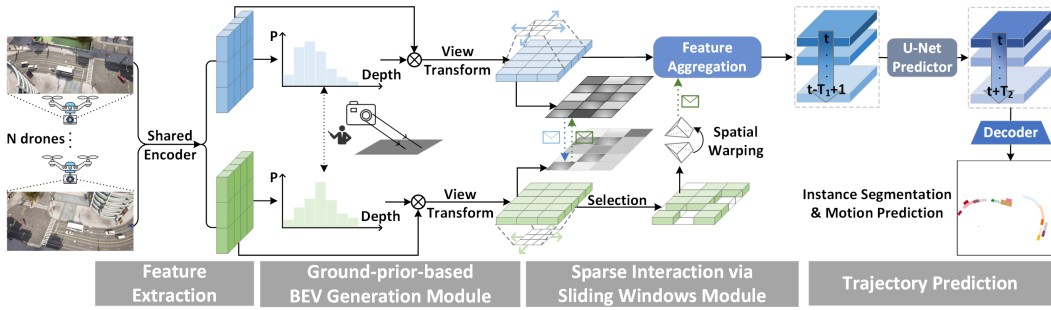

Figure 2: The overall architecture of our proposed DHD framework. For clarity, we just present the collaboration between two drones.

A simulated dataset, "Air-Co-Pred", designed for multi-drone collaborative prediction, is introduced to validate the effectiveness of the proposed DHD framework.

## 2 Related Work

**BEV Generation**. Benefiting from the scale consistency and unified coordinate system, BEV is extensively utilized in collaborative perception. The generation of BEV features is currently categorized into two types: explicit 2D-3D mapping and implicit 3D-2D mapping [16]. In the explicit mapping paradigm, PON [17] introduces a dense transformer module to learn the mappings between image and BEV representations. LSS [13] employs a learnable categorical depth distribution to lift the 2D features to 3D space, and BEVDepth [18] further utilizes explicit supervision on the predicted depth. In the implicit mapping paradigm, BEVFormer [19] leverages the deformable attention mechanism for BEV feature generation, while PolarFormer [17] adopts polar coordinates for precise feature localization. This study adopts a 2D to 3D generation manner similar to LSS. Depth estimation plays a crucial role in the BEV generation, and we are committed to its enhancement.

**Collaborative Strategy.** Current collaborative methods include raw-measurement-based early collaboration, result-based late collaboration, and feature-based intermediate collaboration [20]. Due to the performance-bandwidth trade-off, intermediate collaboration is extensively studied. For example, Who2com [21] introduces a handshake mechanism to select collaborative partners, while When2com [22] further determines when to initiate collaboration. Where2comm [14] employs the detection head to direct regions for sparse interactions. V2VNet [23] achieves multi-round message exchange via graph neural networks. V2X-ViT [24] explore correlations among heterogeneous collaborators through transformer blocks. DiscoNet [25] employs knowledge distillation, whereas CORE [26] utilizes reconstruction concepts to facilitate feature interactions. UMC [27] and SCOPE [28] leverage temporal information to guide feature fusion. However, these strategies are mostly optimized for detection tasks or merely improving feature-level representations. Concerning collaborative forecasting, V2X-Graph [29] utilizes intricate graph structures to forecast motion based on vector maps instead of the vision-based inputs as in this study.

## 3 Methodology

### 3.1 Problem Formulation

This section presents DHD, a well-designed collaborative framework for multi-drone object trajectory prediction, as depicted in Fig. 2. The proposed DHD enables multiple drones to share visual information, promoting more holistic perception and prediction. Conceptually, we consider a scenario with $N$ drones, each capable of sending and receiving collaboration messages from others, and storing $T_1$ historical frames while predicting $T_2$ future trajectories. For the $k$-th drone, $X_k^{t_i}$ denotes the raw observation at input frame $t_i$, and $Y_k^{t_o}$ represents the ground truth at output frame $t_o$. The objective of DHD is to optimize the performance of multi-drone object trajectory prediction under a

total transmission budget $B$:

$$\max \sum_{t_o=1}^{T_2} g\left(\hat{Y}_k^{t_o}, Y_k^{t_o}\right), \quad \text{subject to } \left\{\hat{Y}_k^{t_o}\right\}_{t_o=1}^{T_2} = c_\theta \left\{X_k^{t_i}, \left\{X_{j\to k}^{t_i}\right\}_{j=1}^{N}\right\}_{t_i=1}^{T_1}, \sum_{t_i=1}^{T_1}\sum_{j=1}^{N}\left|X_{j\to k}^{t_i}\right| \le B, \tag{1}$$

where $g(\cdot,\cdot)$ represents the evaluation metric, and $\hat{Y}_k^{t_o}$ is the prediction outcome of drone $k$ at output frame $t_o$. $\mathcal{C}_\theta(\cdot)$ is the collaborative framework parameterized by $\theta$, and $X_{j\to k}^{t_i}$ denotes the message transmitted from the $j$-th drone to the $k$-th drone at input frame $t_i$. The subsequent subsections of Section 3 detail the major components.

## 3.2   2D Feature Extraction of Observations

Initially, each drone captures observations and individually processes its range view (RV) images through a shared encoder $\Phi_{\text{Enc}}(\cdot)$ to extract semantic information. We adopt the backbone of EfficientNet-B4 [30] as the encoder for each drone because of its low inference latency and optimized memory usage, consistent with the literature [31]. For the $k$-th drone, given its observation $X_k^{t_i}$ at the input frame $t_i$, the extracted 2D feature map is denoted as $F_{k,\text{2D}}^{t_i} = \Phi_{\text{Enc}}(X_k^{t_i}) \in \mathbb{R}^{\mathbf{H}\times\mathbf{W}\times\mathbf{C}}$, where $\mathbf{H}$, $\mathbf{W}$, $\mathbf{C}$ denote its height, width and channel, respectively.

## 3.3   Ground-prior-based BEV Generation Module

Depth estimation plays a critical role in generating BEV representations. However, the vanilla LSS method struggles to approximate the precise depth of each pixel due to the vast observation range of drones. Fortunately, the oblique perspective of drones intersects with the ground plane, providing a theoretical upper bound for depth and a nearly zero-altitude reference for each pixel, as shown in Fig. 3. Guided by this principle, we propose the GBG module, which refines depth at each pixel to generate more accurate BEV representations, with its methodology illustrated in Fig. 4. The process below omits the subscript $k$ and $t_i$ for concision.

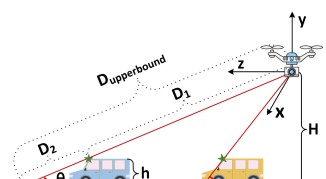

Figure 3: Illustration of the theoretical depth upper-bound and the impact of various viewing angles on depth estimation. The depth of objects, $D_1$, can not exceed $D_{\text{upperbound}}$.

**Derivation of Depth Upper-bound.** Given a drone at altitude $H$, the pinhole camera model is utilized to determine the depth upper-bound $D_{\text{upperbound}}^{(u,v)}$ at each pixel coordinate $(u,v)$, with the help of its camera's intrinsic matrix $\mathbf{K}$, rotation matrix $\mathbf{R}$, and translation vector $\mathbf{T}$:

$$D_{\text{upperbound}}^{(u,v)} = \frac{-H - [\mathbf{R}^{-1}(-\mathbf{T})]_2}{[\mathbf{R}^{-1}\mathbf{K}^{-1}]_{21}u + [\mathbf{R}^{-1}\mathbf{K}^{-1}]_{22}v + [\mathbf{R}^{-1}\mathbf{K}^{-1}]_{23}}. \tag{2}$$

Furthermore, the viewing angle $\theta_{(u,v)}$ of each pixel can be represented as $\arcsin\left(H/D_{\text{upperbound}}^{(u,v)}\right)$.

**Integrating Height into Depth Estimation.** In aerial observations, objects are usually closer to the ground than to the drones. Therefore, it's more feasible to estimate the distance from the objects to the intersection point between their respective viewing ray and the ground, as indicated by line $D_2$ in Fig. 3. Nevertheless, as the viewing angles diminish, the corresponding depth significantly increases. It becomes progressively challenging to estimate the distance to the intersection point due to less visual information available at greater distances. Given the static height of objects and apparent feature discrepancy against the zero-altitude ground plane, it's an alternative to estimate the object's height $h_{\text{pred}}$ and deduce the distance to the intersection points with the help of viewing angles. Accordingly, depth estimation $\Phi_{\text{Depth}}(\cdot)$ can be succinctly expressed as:

$$\Phi_{\text{Depth}}(u,v) = D_{\text{upperbound}}^{(u,v)} - \frac{h_{\text{pred}}}{\sin\left(\theta_{(u,v)}\right)} = D_{\text{upperbound}}^{(u,v)}\left(1 - \frac{h_{\text{pred}}}{H}\right). \tag{3}$$

**BEV Generation Process.** We utilize a parametric network $\Phi_{\text{Height}}(\cdot)$ for height estimation, which generates a pixel-wise categorical distribution of height, represented as $h_{\text{pred}} = \Phi_{\text{Height}}(F_{\text{2D}}) \in \mathbb{R}^{\mathbf{H}\times\mathbf{W}\times\mathbf{D}}$. Here, $\mathbf{D}$ signifies the number of discretized height bins. Subsequently, the estimated height are input into $\Phi_{\text{Depth}}$ to compute the estimated depth $d_{\text{pred}}$. Then, an outer product between

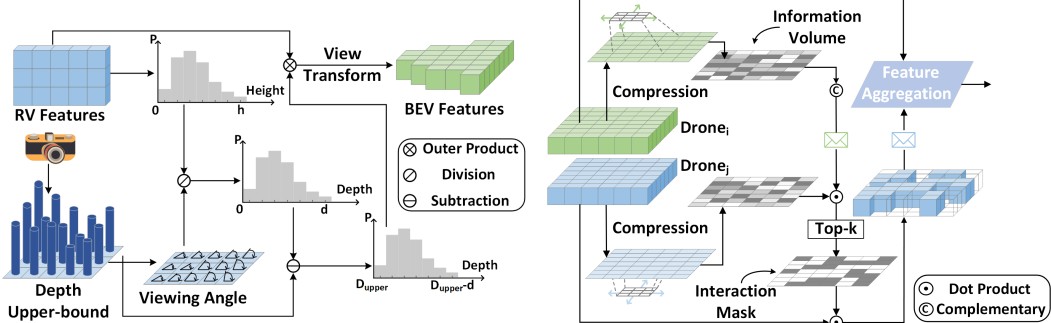

Figure 4: Operation flow of GBG module.  Figure 5: Operation flow of SISW Module.

image features $F_{2D}$ and all potential depth candidates $d_{pred}$ is conducted to generate frustum features, which are then projected into the 3D space based on the inverted pinhole camera model. This process results in 3D voxel representation $V \in \mathbb{R}^{\mathbf{X} \times \mathbf{Y} \times \mathbf{Z} \times \mathbf{C}}$, where $\mathbf{Z}$ refers to the height of voxel. Finally, through sum pooling $\Phi_{Sp}$, these voxel features $V$ are condensed into a single height plane to generate the BEV features: $F_{BEV} = \Phi_{Sp}(V) \in \mathbb{R}^{\mathbf{X} \times \mathbf{Y} \times \mathbf{C}}$, serving for the subsequent collaboration process.

### 3.4 Sparse Interaction via Sliding Windows Module

In aerial observations, objects are typically small and sparsely distributed, leading to critical information occupying only a minor portion of the panoramic view. Given this context, the communication overhead can be considerably reduced by transmitting solely the complementary objects during collaborations [14]. Nonetheless, environmental information is also crucial for prediction tasks, with studies [32, 33] demonstrating that dynamic entities and static textures within surroundings can provide insights for forecasting future trends. Hence, it's necessary to include essential environmental elements within limited data transmissions for better forecasting. To this end, we introduce a novel sparse interaction module named SISW for collaboration among drones. This module assesses the information volume across different areas through sliding windows, determining regions for inter-drone interactions. The SISW module's workflow is illustrated in Fig. 5 and delineated below:

**Information Volume Assessment**. The original BEV features $F_{k,BEV} \in \mathbb{R}^{\mathbf{X} \times \mathbf{Y} \times \mathbf{C}}$ are compressed to $F'_{k,BEV} \in \mathbb{R}^{\mathbf{X} \times \mathbf{Y} \times \mathbf{1}}$ with a trainable $1 \times 1$ convolution, reducing the transmission overhead for inter-drone comparisons of complementary information. Then, a $K \times K$ sliding window traverses over the compressed features, calculating the discrepancy between each position's features and the central features. The calculated discrepancy is then normalized to the interval $(0, 1)$ via a sigmoid function $\sigma$, with the average discrepancy across the window serving as the indicator of the information volume $I$:

$$I_i^{mn} = \frac{1}{K^2} \sum_{a=0}^{K-1} \sum_{b=0}^{K-1} \sigma \left( F'_{i,BEV}(m+a, n+b) - F'_{i,BEV}(m,n) \right), \quad \text{where } (m,n) \in \mathbb{R}^{\mathbf{X} \times \mathbf{Y}}. \quad (4)$$

In this setting, the information volume is almost zero in areas of homogeneity, like empty backgrounds. In contrast, the information volume is much higher in regions with significant feature variations, such as when the area contain the objects or crucial environmental information.

**Selective Sparse Interaction**. Efficient collaboration among drones aims to share complementary information that supplements receivers' current knowledge, thus enhancing the overall system's accuracy. When $k$-th drone receives the information volume from $j$-th drone, complementary scores $S_{j \rightarrow k}$ among drones are calculated as $I_j \odot (1 - I_k)$. To achieve sparse interactions, the interaction mask $M_{j \rightarrow k}$ is determined by the top $K$ from the ranked scores $S_{j \rightarrow k}$ and sparse collaborative features $C_{j \rightarrow k}$ are denoted as $M_{j \rightarrow k} \odot F_{j,BEV}$.

**Collaborative Features Aggregation**. Upon receiving sparse collaborative features, drone $k$ implements a geometric transformation $\mathbf{T}$ to align $C_{j \rightarrow k}$ with its local coordinate system, ensuring spatial congruence. Subsequently, a learnable Gaussian-based interpolation $\phi_{inter}$ is applied to infill undefined values, smoothing the spatial distribution of the features. This step is essential to mitigate the impacts of numerical zero vectors that could adversely affect subsequent feature fusion pro-

cesses [27]. The refined collaborative features are then given by: $C'_{j \to k} = \phi_{\text{inter}}(\mathbf{T}(Z_{j \to k}))$. Guided by local features $F_{k,\text{BEV}}$, the contribution weight $W_j$ of $C'_{j \to k}$ towards constructing aggregated features $A_k$ is quantified by: $W_j = \frac{\phi_{\text{conv}}([F_{k,\text{BEV}};C'_{j \to k}])}{\sum_{j=1}^{N} \phi_{\text{conv}}([F_{k,\text{BEV}};C'_{j \to k}])} \in \mathbb{R}^{\mathbf{X} \times \mathbf{Y} \times \mathbf{1}}$, where $\phi_{\text{conv}}$ represents the multi-layer convolution operations. Moreover, a pixel-level weighted fusion is executed to generate the aggregated features $A_k$ for subsequent downstream tasks: $A_k = \sum_{j=1}^{N} W_j C'_{j \to k} \in \mathbb{R}^{\mathbf{X} \times \mathbf{Y} \times \mathbf{C}}$.

### 3.5 Prediction Decoders and Objective Optimization

Aligned with PowerBEV [34], the state-of-the-art approach for joint perception and prediction, a temporal U-Net architecture is used to interpret a series of $T_1$ aggregated features $\{A_{t_i}\}_{t_i=1}^{T_1}$ and forecast further $T_2$ dynamics, such as instance segmentation $\{y_{t_o}^{\text{seg}}\}_{t_o=1}^{T_2}$ and future flow $\{y_{t_o}^{\text{flow}}\}_{t_o=1}^{T_2}$. For the end-to-end training, we adopt a cross-entropy loss for the segmentation task and a smooth $l_1$ distance for the flow loss. The overall loss function $L$ is formulated as:

$$L = \frac{1}{T_2} \left\{ \sum_{t=0}^{T_2} \gamma^t \left( \lambda_1 L_{\text{ce}}(\hat{y}_t^{\text{seg}}, y_t^{\text{seg}}) + \lambda_2 L_{l_1}(\hat{y}_t^{\text{flow}}, y_t^{\text{flow}}) \right) \right\}, \tag{5}$$

where $\gamma = 0.95$ is a temporal discount parameter and $\lambda_1$, $\lambda_2$ are the balancing coefficients adjusted dynamically through uncertainty weighting. In the inference phase, Hungarian algorithm correlates multi-frame instances to synthesize present instance segmentation and future motion prediction.

## 4 Experiments

### 4.1 Datasets

The available datasets for multi-drone collaboration [14] primarily focus on detection and segmentation, lacking the support for the prediction tasks. To bridge this gap, we create a simulated dataset "Air-Co-Pred" for collaborative trajectory prediction based on CARLA[15]. Specifically, four collaborative drones are positioned at intersections to monitor traffic flow from different directions. These drones fly at an altitude of 50 meters, covering an area of approximately 100m $\times$ 100m. They capture images at a frequency of 2Hz to support the temporal prediction task. The collected dataset includes 32k synchronous images with a resolution of $1600 \times 900$ and is split into 170 training scenes and 30 validation scenes. Each frame is well-annotated with both 2D and 3D labels, comprising three major object categories: vehicles, cycles, and pedestrians. Given the challenges of tiny objects from aerial perspectives, this study mainly concentrates on the vehicle category, which includes many sub-categories to augment the robustness of identifying various vehicles. To illustrate the challenges of aerial observations intuitively, we present several charts to reflect the characteristics of "Air-Co-Pred", such as occlusion, long-distant observations, small objects, etc., as shown in Fig.6.

### 4.2 Evaluation Metrics

To evaluate the localization accuracy of generated BEV representations, we utilize the metrics of precision and recall to reflect matches between predicted and ground-truth instance centers. Additionally, the L2 distance is used to quantify the position deviations for successfully matched instances. Regarding downstream tasks, we adopt two metrics widely used in previous works [31, 34]. For frame-level evaluation, Intersection-over-Union (IoU) evaluates the segmentation quality of objects at the present and future frames. The specific calculation is as follows:

$$\text{IoU}(\hat{y}_t^{\text{seg}}, y_t^{\text{seg}}) = \frac{1}{N} \sum_{t=0}^{N-1} \frac{\sum_{h,w} \hat{y}_t^{\text{seg}} \cdot y_t^{\text{seg}}}{\sum_{h,w} \hat{y}_t^{\text{seg}} + y_t^{\text{seg}} - \hat{y}_t^{\text{seg}} \cdot y_t^{\text{seg}}}, \tag{6}$$

where $N$ denotes the number of output frames, $\hat{y}_t^{\text{seg}}$ and $y_t^{\text{seg}}$ denote the predicted and ground truth semantic segmentation at timestamp $t$, respectively. As for the video-level evaluation, Video Panoptic Quality (VPQ) reflects the quality of the segmentation and ID consistency of the instances through the video, expressed as:

$$\text{VPQ}\left(\hat{y}_t^{\text{inst}}, y_t^{\text{inst}}\right) = \sum_{t=0}^{N-1} \frac{\sum_{(p_t, q_t) \in TP_t} \text{IoU}(p_t, q_t)}{|TP_t| + \frac{1}{2}|FP_t| + \frac{1}{2}|FN_t|}, \tag{7}$$

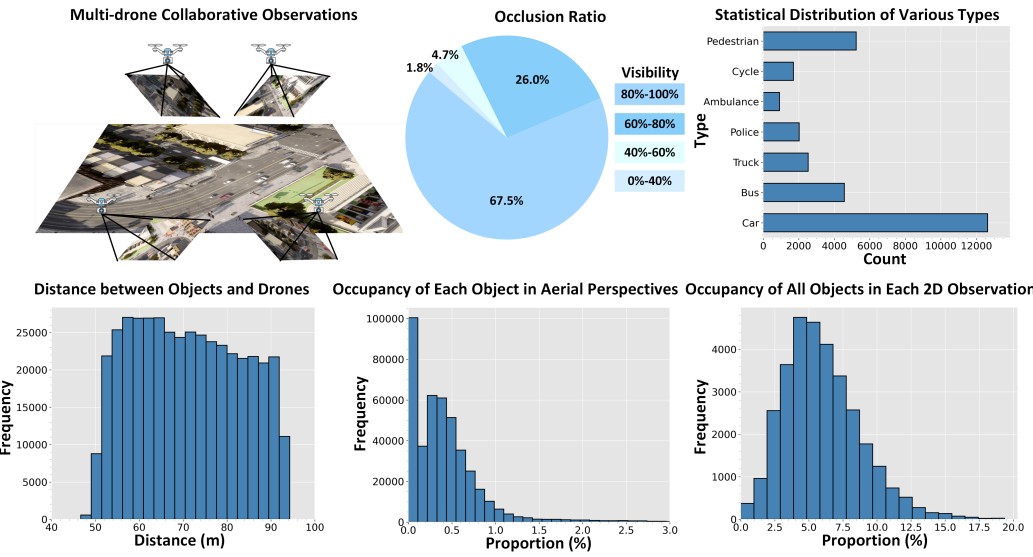

Figure 6: Statistical charts for the Air-Co-Pred dataset, depicting the occlusion within a single view, the number of various object types, the distribution of distances between objects and drones, and the proportion of objects within observations, respectively.

Table 1: An analysis of trajectory prediction and localization error across different BEV generation baselines. They follow the SISW-based interaction mechanism for subsequent multi-drone collaboration. DHD (w/o H) denotes the DHD variant without integrating height estimation.

| Models | IoU (%) ↑ | | VPQ (%) ↑ | | Precision (%) ↑ | | Recall (%) ↑ | | Deviation (m) ↓ | |
|---|---|---|---|---|---|---|---|---|---|---|
| | Short | Long | Short | Long | Short | Long | Short | Long | Short | Long |
| LSS [13] | 56.29 | 51.76 | 44.98 | 43.13 | 70.97 | 75.68 | 77.88 | 80.49 | 1.08 | 2.18 |
| DVDET [35] | 59.37 | 48.91 | 49.98 | 41.84 | 77.59 | 56.62 | 77.85 | **84.24** | 1.00 | 2.28 |
| DHD (w/o H) | 58.28 | 52.73 | 48.01 | 44.37 | **83.37** | 76.21 | 77.14 | 81.28 | 1.07 | 1.67 |
| DHD (Ours) | **61.39** | **53.95** | **50.43** | **46.15** | 81.47 | **85.45** | 80.25 | 77.98 | **0.82** | **1.46** |

where $TP_t, FP_t$, and $FN_t$ correspond to true positives, false positives, and false negatives at timestamp $t$, respectively. True positives are predicted instances with the IoU greater than 0.5 and consistent instance IDs with the ground truth, whereas false positives and false negatives are incorrectly predicted and missed instances, respectively.

## 4.3 Implementation Details

We follow the setup from the existing study [10] for collaborative trajectory prediction. Initially, the raw images, with a resolution of $900 \times 1600$ pixels, are scaled and cropped to a size of $224 \times 480$. As to view transformation, the height estimation range is configured from 0 to 10 meters, discretized into 100 intervals. Subsequently, for BEV representations, the spatial ranges for the $x$, $y$, and $z$ axes are set to $[-50, 50]$, $[-50, 50]$, and $[-60, -40]$ meters, respectively. We evaluate the model performance across various perceptual scopes: a 100 m×100 m area with 0.5 m resolution (long) and a 50 m×50 m area with 0.25 m resolution (short). Temporally, we utilize three frames from the past 1s (including the present frame) to predict both semantic segmentation and instance motion for the future four frames (2s). Besides, we select a 7×7 window size and set the transmission ratio as 25% for the SISW module for comprehensively optimal performance. The relative details on hyper-parameter ablation studies are provided in the supplementary material. Our DHD framework is trained with Adam optimizer at an initial learning rate of $3 \times 10^{-4}$. It runs on four RTX 4090 GPUs, handling a batch size of 4 for 20 epochs.

Table 2: A comparison of collaboration baselines for enhanced prediction performance. **Early collaboration** refers to the collaboration of raw observations, where multi-view images jointly generate BEV representations. **Intermediate collaboration** focuses on the feature-level interactions to achieve comprehensive BEV representations. The fully connected paradigm shares complete features among all members, while the partially connected one restricts interactions to certain members or regions. **Late collaboration** merges the individual prediction results from multiple drones. All the collaboration approaches adopt the GBG module for BEV generation. Our DHD performs best in the partially connected intermediate collaboration paradigm.

| Models | | IoU (%) ↑ | | VPQ (%) ↑ | |
|---|---|---|---|---|---|
| | | Short | Long | Short | Long |
| No Collaboration | | 38.1 | 32.6 | 31.1 | 27.8 |
| Early Collaboration | | 62.8 | 54.5 | 51.7 | 45.9 |
| Late Collaboration | | 57.1 | 51.4 | 47.5 | 43.6 |
| Intermediate Collaboration (Fully Connected) | V2X-ViT [24] | 61.5 | 53.3 | 50.2 | 45.7 |
| | V2VNet [23] | 59.6 | 53.8 | 50.5 | 46.9 |
| Intermediate Collaboration (Partially Connected) | Who2com [21] | 52.9 | 44.3 | 40.5 | 37.0 |
| | When2com [22] | 55.7 | 45.8 | 40.8 | 40.4 |
| | Where2comm [14] | 57.5 | 51.4 | 48.6 | 44.2 |
| | UMC [27] | 57.3 | 52.3 | 48.3 | 44.3 |
| | UMC (w/o GRU) | 58.8 | 53.2 | 48.5 | 45.9 |
| | DHD (Ours) | **61.4** | **54.0** | **50.4** | **46.2** |

## 4.4 Quantitative Evaluation

**Benchmark Comparison in BEV Generation.** We select the vanilla LSS [13] and the drone-specific DVDET [35] as baselines. For fairness, their depth estimation range is set from 1 to 100 meters, divided into 100 intervals. As shown in Table 1, DHD outperforms the vanilla LSS regarding downstream performance, showing a 9.06% improvement in IoU and a 12.11% increase in VPQ within short-range observations. Furthermore, it demonstrates a 5.06% improvement in IoU and a 6.42% increase in VPQ for the long-range setting. While DVDET incorporates a deformable attention mechanism to refine BEV representations, it exhibits gains in the short-range setting but a notable decline in the long-range setting in contrast to the vanilla LSS. Inaccurate depth estimation causes the projected objects to shift from their correct positions, resulting in mismatches with the ground truth. Specifically, DHD achieves fewer mistaken and missing matches and better localization, with over a 20% reduction in position deviation. Remarkably, the DHD without height estimation still achieves performance gains and reduces long-range localization errors solely by relying on ground priors.

**Benchmark Comparison in Collaboration.** Table 2 shows that our DHD achieves performance comparable to early collaboration, particularly within the long perceptual range setting. It also significantly outperforms No-Collaboration, with an increase of 42.61% in IoU and 45.90% in VPQ for the short-range setting, and 74.29% in IoU and 79.36% in VPQ for the long-range setting, respectively, revealing the effectiveness of collaboration. In contrast to fully connected baselines, DHD can approach and even exceed similar performance with only a quarter of the transmission ratio. When compared to the partially connected baselines, DHD surpasses When2com, the upgraded version of Who2com, by almost 20% in both IoU and VPQ. Furthermore, DHD also beats the previous state-of-the-art method, Where2com, by a considerable margin, for example, 5.82% IoU and 3.85% VPQ in the long-range setting. Because our DHD not only considers the foreground objects but also incorporates relevant environmental information for predictions. Notably, the recent state-of-the-art method, UMC, with GRU-based feature fusion, exhibits 3∼4% lower performance than ours. We find that its temporal fusion deteriorates original downstream tasks. Although this temporal prior has enhanced object detection, it may lead to perception aliasing for prediction tasks.

## 4.5 Qualitative Evaluation

**Visualization of Prediction Results.** Fig. 7 demonstrates that collaboration among drones, when compared to no collaboration, facilitates the acquisition of the location and state of occluded and out-of-range objects through feature-level interactions. Furthermore, DHD correctly forecasts the trajectory of multiple objects in challenging intersections and achieves more accurate segmentation

and prediction results compared to the well-known baseline, Where2com. This is attributed to Where2comm's transmitted features to focus on foreground objects while neglecting the surroundings that contributed to downstream tasks. These findings are consistent with our quantitative evaluations.

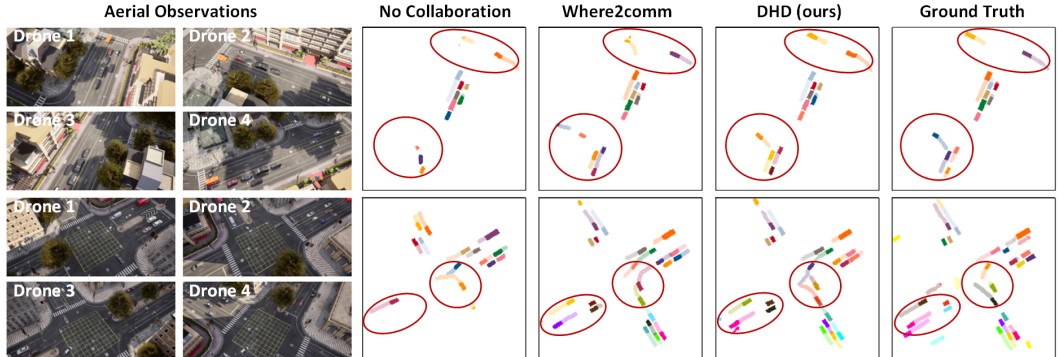

Figure 7: A comparative analysis of visualizations among various collaboration baselines. Each instance is allocated a distinct color, and its predicted trajectory is represented with the same color and slight transparency. Red circles highlight the areas where other baselines make mistaken predictions.

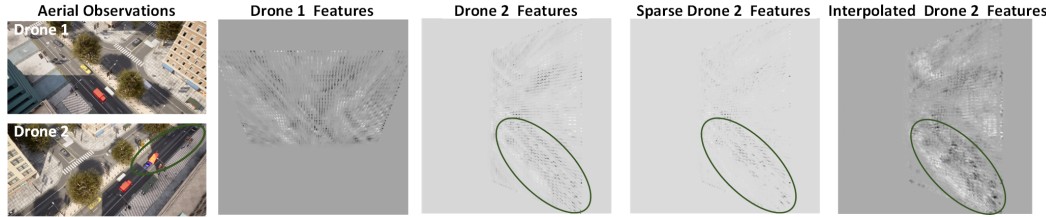

Figure 8: The visualization of feature-level interactions. Green circles represent the primary complementary area, including missing objects and crucial environmental cues.

**Visualization of Sparse Interactions.** To examine the interaction during the collaboration, we analyze sparse features transmitted between Drone 1 and Drone 2 via the SISW module. Fig.8 shows that transmissions include not only object features (in dark gray) but also surrounding features (in light white), which is more prominent in column 5. Columns 2 and 3 highlight an obvious discrepancy in the mean of features, which could affect subsequent fusion processes. We implement a learnable interpolation to fill in blank features to alleviate the gap. Above all, these visualizations are in agreement with our designs and expectations.

### 4.6 Ablation Studies

**Effectiveness of Proposed Modules.** Our DHD framework introduces two innovative components: the GBG and SISW modules. We evaluate these modules based on their ability to enhance prediction accuracy and optimize the performance-transmission trade-off, as depicted in Table 3. The GBG-only variant significantly enhances prediction accuracy with improvements of approximately 10% in short-range and 5% in long-range predictions, primarily attributable to the more accurate BEV representations guided by the ground prior. The SISW-only variant reduces transmission costs by 75% with only a marginal performance decrease of about 1% relative to the baseline. Overall, our DHD, integrated with both modules, can achieve a balance between prediction enhancement and transmission cost.

Table 3: The ablation study of proposed modules. 'Ratio' in the table refers to the transmission ratio during collaborative interactions.

| GBG | SISW | IoU (%) ↑ | | VPQ (%) ↑ | | Ratio |
|---|---|---|---|---|---|---|
| | | Short | Long | Short | Long | |
| – | – | 57.1 | 52.1 | 45.5 | 43.5 | 1 |
| ✓ | – | 61.8 | 54.5 | 51.7 | 45.9 | 1 |
| – | ✓ | 56.3 | 51.8 | 45.0 | 43.1 | 0.25 |
| ✓ | ✓ | 61.4 | 54.0 | 50.4 | 46.2 | 0.25 |

**Performance Changes with the Number of Collaborative Drones.** To explore how the number of collaborating drones affects performance, we conducted relevant experiments, as shown

in Table 4. For the short-range area of 50m × 50m, three drones are sufficient to predict trajectories, with performance comparable to that of four drones. However, for the long-range area of 100m × 100m, predictive performance improves as the number of drones increases. This difference can be attributed to the positional layout of the drones in the dataset, Air-Co-Pred. The drones are situated near intersections to monitor traffic flow, facilitating comprehensive coverage of the short-range areas close to these intersections. In contrast, much of the information in the long-range areas extends along specific road branches, which may only be captured by a single drone. Therefore, increasing the number of drones results in more comprehensive coverage of the long-range areas.

Table 4: Performance metrics for different numbers of drones.

| Num of Drones | IoU (%) ↑ | | VPQ (%) ↑ | |
|---|---|---|---|---|
| | Short | Long | Short | Long |
| 1 | 41.5 | 31.1 | 33.5 | 25.6 |
| 2 | 57.1 | 45.4 | 48.2 | 38.4 |
| 3 | 61.3 | 50.2 | 50.4 | 43.4 |
| 4 | 61.4 | 54.0 | 50.4 | 46.2 |

## 4.7 Generalization to Collaborative 3D Object Detection

We also conduct generalization validation for collaborative 3D object detection in CoPerception-UAVs [14], a publicly available multi-drone collaborative dataset. We select several models for BEV generation baselines: BEVDet [36] (a modified LSS model for detection), its temporal version BEVDet4D [37], and DVDET [35], the official detector for CoPerceptionUAVs. All of these models adopt the SISW module for inter-drone feature-level interactions. The evaluation metrics[38] include mean Average Precision (mAP), mean Absolute Trajectory Error (mATE), mean Absolute Scale Error (mASE), and mean Absolute Orientation Error (mAOE), representing detection accuracy, offset error, size error, and orientation error, respectively. As illustrated in Table. 5, our DHD achieves the best performance in mAP, mATE, and mASE. Specifically, DHD shows a 25.2% improvement in mAP and reductions of 13.7% in mATE and 2.9% in mASE compared to BEVDet. Although BEVDet4D refines depth estimation with temporal information, the results indicate that the ground prior is more critical for aerial depth estimation. Notably, DVDET outperforms DHD in orientation error, probably attributed to its deformable attention.

## 5 Conclusion and Limitation

This paper presents DHD, a collaborative framework for multi-drone object trajectory prediction. Its GBG module leverages the ground prior and simpler height estimation for more accurate BEV representations. Meanwhile, the SISW module adaptively selects regions for collaborative interactions, guided by the sliding window's information volume calculation. Additionally, we construct the first simulated dataset of multi-drone collaborative prediction, named "Air-Co-Pred", to evaluate the effectiveness of DHD through quantitative and qualitative experiments.

Table 5: A comparative analysis of collaborative 3D object detection across different BEV generation baselines.

| Models | mAP ↑ | mATE ↓ | mASE ↓ | mAOE ↓ |
|---|---|---|---|---|
| BEVDet [36] | 0.349 | 1.011 | 0.171 | 1.601 |
| BEVDet4D [37] | 0.371 | 0.949 | 0.172 | 1.118 |
| DVDET [35] | 0.387 | 0.904 | 0.170 | **0.844** |
| DHD (ours) | **0.437** | **0.872** | **0.166** | 0.869 |

**Limitation and Future Work.** The current work only exploits the simulated setting, an idealized scenario, for multi-drone object trajectory prediction. For practical applicability, future efforts will extend to real-world environments, taking into account realistic challenges such as flight turbulence, rough terrain, camera extrinsic noise, latency, communication frequency etc. Specifically, flight turbulence and rough terrain can impact the derivation of depth upper-bound in the GBG module, while inaccurate camera parameters may compromise multi-view projection transformations. Additionally, effectively managing latency and optimizing communication frequency are crucial for collaboration within the SISW module. Addressing these issues is essential for deploying the proposed model in real-world scenarios.

**Acknowledgement.** This work was supported by the National Nature Science Foundation of China under Grant 62331027, and supported by the Strategic Priority Research Program of the Chinese Academy of Sciences, Grant No. XDA0360303.

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

# 6 Appendix

## 6.1 Air-Co-Pred Dataset Details

**Map Creation.** The simulation scenes, including the road layouts, static objects, and traffic flow, are generated using the CARLA simulation platform. We utilize the "Town10" map provided by CARLA as the foundational road layout. This scenario is similar to the intelligent urban setting, with high-rise buildings and roadside trees leading to observation occlusions. These challenges can be effectively addressed through multi-drone collaboration.

**Traffic Flow Creation.** Moving vehicles in the scene are controlled via CARLA, with hundreds of vehicles spawned in each scene using the official script provided by CARLA. The map's road layout determines each vehicle's initial location and motion trajectory.

**Sensor Setup.** Each drone is equipped with a forward-facing RGB camera with a 90° field of view and a resolution of $1600 \times 900$. The cameras are fixed, with their internal positions and rotation degrees remaining constant. During data collection, each camera's translation (x, y, z) and rotation (w, x, y, z in quaternion) are recorded in global and ego coordinates. With this sensor configuration, four collaborative drones flying at a height of 50 meters can effectively cover an area of 100m × 100m.

**Data Collection.** Our proposed dataset is collected by the CARLA simulation platform under the MIT license. We utilize CARLA to create complex simulation scenes and traffic flow. Aerial observation samples are collected at a frequency of 2 Hz. We synchronously collect images from four drones, resulting in four images per sample. Additionally, camera intrinsics and extrinsics in global coordinates are provided to support coordinate transformation. A total of 32,000 images have been collected to support our experiments. Our ground truth labels for collaborative prediction are derived from 3D bounding boxes of observed targets. Thus, we take advantage of lidar sensors to collect these 3D bounding boxes, which include location (x, y, z), rotation (represented with quaternion), and dimensions (length, width and height), amounting to nearly 430,000 3D bounding boxes.

**Data Usage.** We randomly split the samples into training and validation sets, yielding 25.6k images for training and 4.4k for validation. The dataset is structured similarly to the widely used autonomous driving dataset, nuScenes, so that it can be directly used with the well-established nuScenes-devkit.

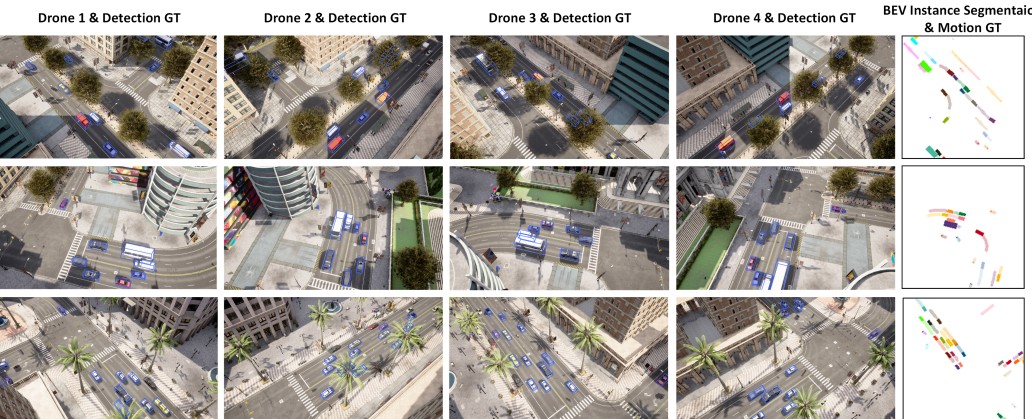

Figure 9: The visualization of our Air-Co-Pred dataset along with the detailed annotations for collaborative detection, segmentation and prediction tasks.

## 6.2 Comparison with Other Existing Datasets

To demonstrate the contributions of our dataset, Air-Co-pred, we conduct a comparative analysis of existing datasets in the multi-drone collaboration domain. By the submission deadline, several datasets are available for multi-drone collaboration, including two real-world datasets (VisDrone-MDOT [2] and VisDrone-MDMT [1]) and two simulation datasets (Airsim-Map [22] and CoPerception-UAVs [14]).

Regarding the existing real-world datasets, the VisDrone series has collected a substantial amount of real-world video data. However, these datasets are constructed solely from a visual perspective and do not provide any information about the drones' poses or camera parameters. As a result, they are limited to supporting 2D visual algorithms [39] such as ReID and object tracking, and cannot be used to evaluate our proposed collaborative prediction framework, which integrates both visual and spatial information.

The simulation dataset, Airsim-Map, is designed to demonstrate the effectiveness of who2com and when2com in mitigating image degradation. However, it only provides multi-view 2D semantic segmentation masks. The dataset most similar to ours is CoPerception-UAVs, proposed by where2comm. While this dataset focuses exclusively on multi-drone collaborative 3D object detection and has been used to validate frameworks like Where2comm [14] and CoCa3D [40], it falls short in addressing joint temporal tasks. Additionally, its large sampling intervals are inadequate for validating our DHD. To bridge this gap, we propose a more comprehensive dataset that includes detailed annotations and an appropriate sampling frequency. This dataset supports a wide range of tasks, including 2D/3D detection [41–43], BEV segmentation [31], multi-object tracking [44], and trajectory prediction [34], and facilitates the preliminary validation of multi-drone collaboration across various scenarios within a simulation environment.

## 6.3 Theoretical Justification and Derivation of the GBG Module

**The Advantage of Depth Estimations from a Near-Ground Perspective**. In aerial observations, objects are generally closer to the ground rather than to the drones, which introduces particular considerations for depth estimation. The estimated range determined by the distance to the drone is represented by $[H - h_{max}, D^{(u,v)}_{\text{upperbound}}]$, where $H$ denotes the altitude of the drone, $h_{\max}$ represents the maximum height of the objects, and $D_{\text{upperbound}}(u, v)$ corresponds to the depth upper-bound of pixel $(u, v)$. In contrast, considering the distance to the ground, the required range is $\left[h_{max}, \frac{h_{max}}{\sin(\theta_{\text{lb}})}\right]$, where $\theta_{\text{lb}}$ denotes the lower-bound of the viewing angle. The discrepancy between these two ranges is significant:

$$\frac{D^{(u,v)}_{\text{upperbound}} - H + h_{max}}{\frac{h_{max}}{\sin\theta_{lb}} - h_{max}} = \frac{\frac{kh_{max}}{\sin\theta_{(u,v)}} - kh_{max} + h_{max}}{\frac{h_{max}}{\sin\theta_{lb}} - h_{max}} = \frac{\frac{k}{\sin\theta_{(u,v)}} - k + 1}{\frac{1}{\sin\theta_{lb}} - 1} \gg 1. \tag{8}$$

Here we substitute $D_{(u,v)}$ with $H$ and set $H = k \cdot h_{max}$, where $k \gg 1$. Depth estimation as an n-class problem becomes increasingly complex as the interval range expands. Therefore, it is more feasible to estimate the distance from objects to the point where their sight intersects the ground.

**Rationality for Height-Based Depth Estimation**. As a car of height $h$ moves away from the camera, its viewing angle $\theta_{(u,v)}$ diminishes, as depicted in Fig. 3. A small angular change $\delta$ incurs a depth variation $\Delta d$, given by:

$$\Delta d = \frac{h}{\sin(\theta_{(u,v)} - \delta)} - \frac{h}{\sin\theta_{(u,v)}} = h\frac{\cos\theta_{(u,v)}}{\sin^2\theta_{(u,v)}}\delta. \tag{9}$$

Additionally, $\Delta d$ is a monotonically decreasing function of the angle $\theta$. As $\theta$ decreases, the corresponding depth significantly increases, which exacerbates the challenge of depth estimation due to less visual information available at greater distances. In contrast, an object's height remains constant regardless of its position, making it relatively easier to estimate. Furthermore, the task of height estimation can even be simplified to an object classification task. For instance, the ground plane maintains an altitude of zero, while different types of vehicles are associated with specific height values.

## 6.4 Additional Qualitative Evaluation

**Impact of Flight Turbulence and Rough Terrain on DHD Performance.** We acknowledge that flight vibrations and uneven terrain can interfere with the drone's relative height to the ground, affecting the BEV generation from the GBG module. Therefore, we introduce perturbations to the drone's altitude to simulate these conditions.

Specifically, we introduce Gaussian noise to the drone's altitude, with noise levels ranging from 0.002 to 0.01. At the highest level, this results in a maximum altitude variation of 0.5 meters, which

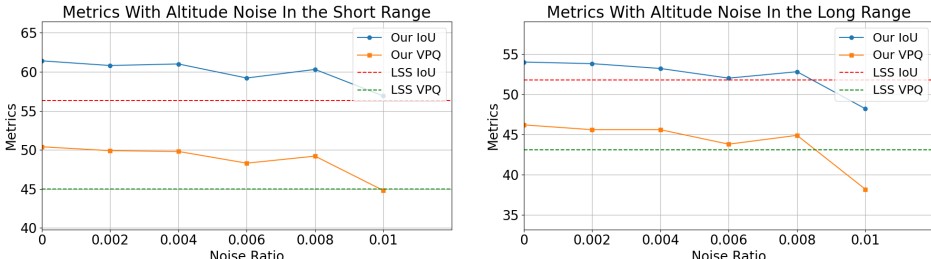

Figure 10: Impact of introducing varying levels of altitude noise on collaborative prediction performance. The dashed lines represent the performance of the depth-based baseline, LSS.

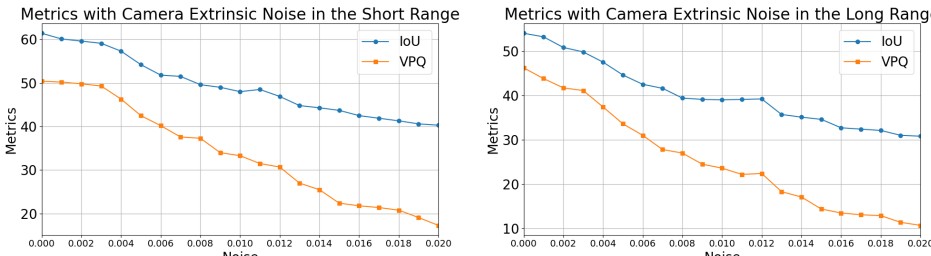

Figure 11: Impact of varying noise levels in camera extrinsics on collaborative prediction performance.

is significant for drone flight. As illustrated in Fig. 10, the results demonstrate that when the noise level exceeds 0.01, the depth estimation advantage conferred by the geometric prior in the GBG module diminishes. However, this limitation of terrain variations is not unique to our GBG module. Mainstream BEV generation methods, such as LSS, BEVFormer, and PETR, also assume a flat ground. Complex terrain requires further study and can be considered a distinct research direction.

The primary goal of our GBG is to explore BEV generation specifically designed for drones. While this is an initial attempt, we recognize the need to account for more complex real-world conditions in future deployments. To address this, we propose a simple yet potentially effective solution: developing a ground flatness estimation module to assess variations in the ground plane, allowing the estimated object height to be adaptively adjusted and thereby mitigating the impact of uneven terrain on subsequent BEV generation.

**Impact of Sensor Noise on DHD Performance.** Sensor noise primarily affects the accurate acquisition of camera extrinsic parameters. To examine this effect, we introduce Gaussian noise to simulate varying levels of disturbance in the extrinsic calibration.

As illustrated in Fig. 11, increasing noise results in a gradual decline in both IoU and VPQ. In short-range settings, noise ratios below 0.003 cause negligible performance drops. However, IoU decreases by about 25% when the noise ratio is between 0.003 and 0.013. Between 0.013 and 0.020, the decline slows, with an additional reduction of approximately 10%. VPQ exhibits a similar trend. In long-range settings, noise ratios below 0.003 also result in acceptable performance declines. However, when the ratio reaches 0.005, noticeable performance degradation occurs, with IoU dropping by 21.3% and VPQ by 32.9%. Overall, noise has a more pronounced impact on VPQ than on IoU, indicating that camera extrinsic bias more severely affects the consistency of future trajectory predictions. Furthermore, the greater impact of extrinsic noise in long-range observations can be attributed to objects at long distances often being observed from a single perspective, lacking the multi-view validation available in short-range scenarios.

These results demonstrate that our DHD can tolerate a small amount of sensor-based extrinsic noise. Besides, larger biases in extrinsic parameters can significantly impact collaborative prediction. Therefore, accurate estimation of these parameters is crucial for maintaining high performance in collaborative perception systems. This finding is equally applicable to real-world scenarios.

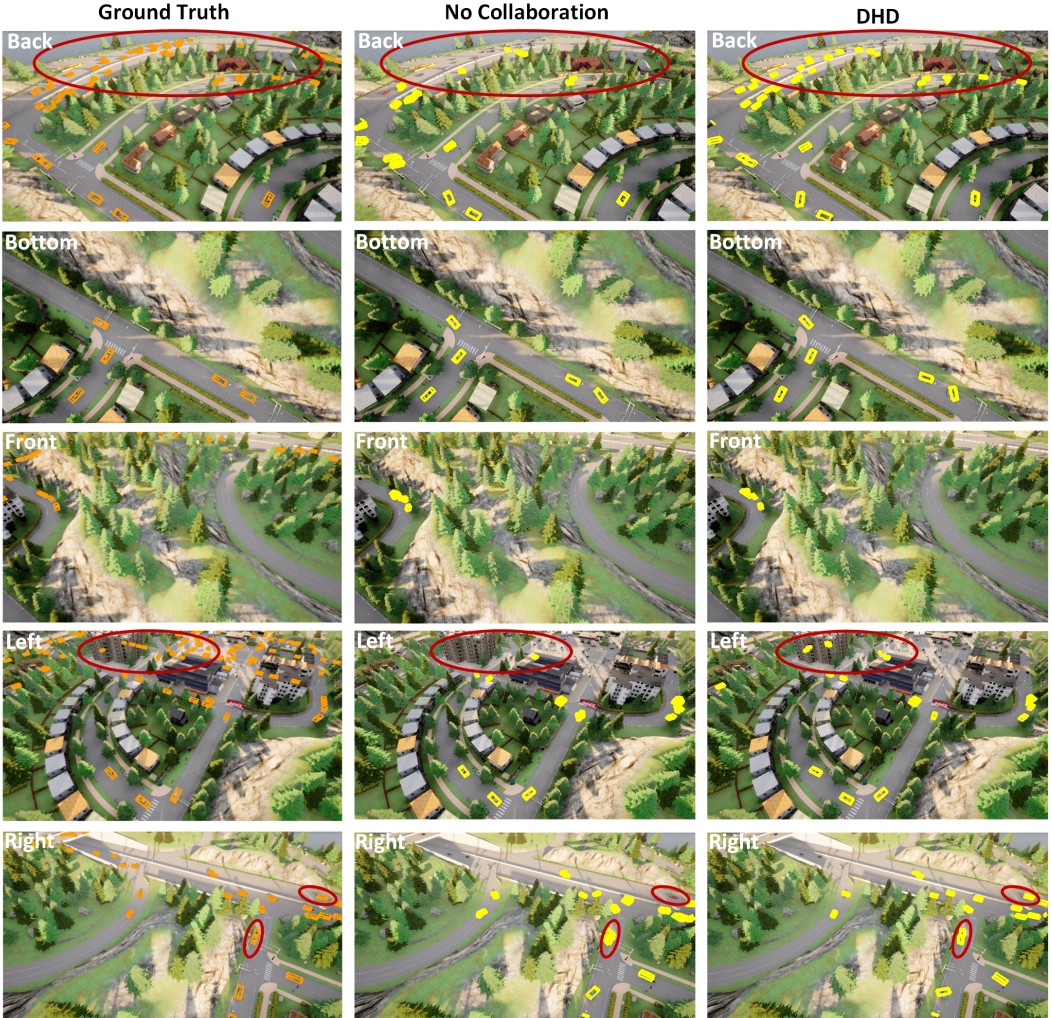

Figure 12: The visualizations illustrate the enhancement of collaborative 3D object detection in CoPerception-UAVs. For clarity, we select one of the five collaborative drones in this dataset, each equipped with five cameras. The red circles highlight areas where single-drone perception fails to detect objects, but our DHD framework successfully detects them.

**Visualization of Collaborative 3D Object Detection.** As illustrated in Fig. 12, although our DHD framework is not specially designed for collaborative 3D object detection, it still effectively overcomes the limitations of single-drone perception, such as occlusion and out-of-range observations, through multi-drone collaboration.

### 6.5 Ablation Studies on Hyper-Parameters

**Effect of Sliding Window Size in SISW Module** The size of the window determines the contextual range around the central point. Smaller windows are more sensitive to capturing high-frequency local feature variations, but they may also be susceptible to noise. Conversely, larger windows encompass a broader range of information but potentially neglect valuable local discrepancies. After investigating the effects of various window sizes, as shown in Fig. 13, we select a $7 \times 7$ window size to assess the information volume in the SISW module for optimal performance.

**Performance Enhancement Brought by Transmission Ratio.** Fig. 8 shows that each drone's effective BEV feature coverage is approximately 50%. Accordingly, our transmission ratio decreases from 50% to 0.1% to explore the balance between performance and bandwidth. In the curve shown in Fig. 13, as the ratio decreases from 50% to 25%, there is negligible performance variation, maintaining

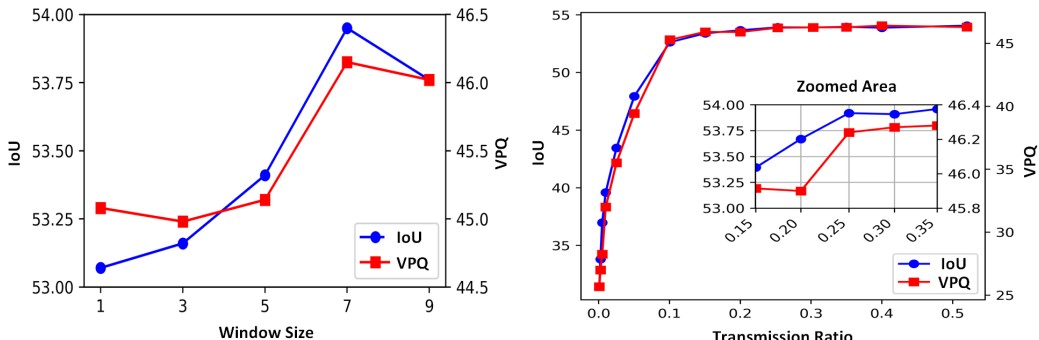

Figure 13: Collaborative prediction performance of DHD in long-range settings with the varied transmission ratio and sliding windows.

a relatively high value. However, a significant decline in performance occurs below 10%, with no discernible benefit below 0.25%. To consider comprehensively, we set the transmission ratio as 25% for DHD's SISW module.

