# OpenReview forum: "Drones Help Drones: A Collaborative Framework for Multi-Drone Object Trajectory Prediction and Beyond"
_NeurIPS.cc/2024/Conference — NeurIPS 2024 poster_

### Official Review · Reviewer_vjyF · 2024-07-07

**Soundness:** 4
**Presentation:** 3
**Contribution:** 3
**Rating:** 7
**Confidence:** 5

**Summary:**

This paper presents a collaborative framework for multi-drone object trajectory prediction, named DHD, which consists of two specifically designed components: the GBG module, aimed at generating more accurate BEV representations in aerial scenes, and the SISW module, which adaptively selects regions for collaborative interactions. Additionally, a simulated multi-drone collaborative observation dataset is created to demonstrate the effectiveness of the proposed DHD framework.

**Strengths:**

- This paper successfully implements an end-to-end multi-drone object trajectory prediction based purely on visual inputs and extends it to the collaborative object detection task. This work fills in the blank of end-to-end multi-drone collaborative prediction, demonstrating notable originality.
- The proposed GBG module innovatively leverages the unique inclined observations from drone perspectives as geometric priors and replaces traditional depth estimation with height estimation. These enhancements significantly improve the accuracy of BEV representations for long-range aerial observations compared to the listed baselines, such as LSS and DVDET.
- This paper presents an efficient communication strategy called SISW, which considers both the limited inter-drone communication and the prediction task's dependency on foreground and contextual information. Experimental results indicate that this design outperforms previous sparse interaction strategies like where2comm.
- A simulated dataset, "Air-Co-Pred," is created to support multi-drone object trajectory prediction. The dataset enriches the available resources within the multi-drone field and can serve as a valuable benchmark for future research.

**Weaknesses:**

- The authors need to carefully review the paper for grammar mistakes, typos, and formatting errors. Specific issues include:
    - Line 82: BEV's full-term appears redundantly.
    - Line 119: The subscript in$ Y_{k}^{o} $ is incorrect and should be$ Y_{k}^{t_o} $.
    - Line 152: Figure reference is incorrect.
    - Line 221: "2HZ" is improperly formatted and should be "2Hz."
    - Equation 6: The subscript next to $ \Sigma $ should be $ xy $ instead of $ hw $.
Addressing these issues will improve the paper's readability.

- The transmission volumes for various collaboration strategies in Table 2 are unclear. It would be beneficial to supplement this information to better reflect the balance between the performance improvements in prediction and the cost of collaborative interactions. Additionally, it appears that the results of DHD in Table 2 are not the best in every column, so why are they bolded?
- Please consider exploring scenarios where the number of drones varies from 1 to 4. This would help investigate the impact of drone quantity on perception enhancement and also reflect the influence of potential drone failures on performance metrics.
- In reality, extrinsic parameters are derived through computation and approximation, leading to certain biases. Please explore how the extent of extrinsic bias affects the performance of collaborative prediction.

**Questions:**

- Could you provide more details about Air-Co-Pred, including the drone attitude settings, flight patterns, the speed of objects, etc.? Why were four drones chosen to collect data?
- In Section 3.3, the equation regarding the derivation of the depth upper-bound is presented directly without any intermediate steps in the main text or supplementary materials. Please provide the detailed derivation process to help readers understand it better.
- The reviewer found that the proposed GBG module is highly relevant to the applications of camera intrinsics and extrinsic in UAVs. The reviewer encourages the authors to acknowledge the existence of related works and clarify the differences.
For reference, see:
Shen H, Lin D, Yang X, et al. Vision-Based Multi-Object Tracking through UAV Swarm[J]. IEEE Geoscience and Remote Sensing Letters, 2023.
Pan T, Dong H, Deng B, et al. Robust Cross-Drone Multi-Target Association Using 3D Spatial Consistency[J]. IEEE Signal Processing Letters, 2023.
- Considering the complexity of the proposed framework, could you provide additional details or code for better understanding and reproducibility?

**Limitations:**

The authors acknowledge that the current focus of this paper is on simulated environments to investigate multi-drone object trajectory prediction without considering latency and camera noise. These are reasonable limitations, and addressing them in future work would be beneficial

---

> ### Author Rebuttal · Authors · 2024-08-06
>
> ### [Weakness 1]: Grammar mistakes
>  We have carefully reviewed and corrected the grammar mistakes, typos, and formatting errors.
> ### [Weakness 2]: Transmission volumes. Why the results of DHD in Table 2 bold?
> Based on the V2V communication protocol [1], data broadcasting can achieve a bandwidth of 27 Mbps at a range of 300 meters. Considering the capture frequency of 2 Hz, the bandwidth requirement should be less than 13.5 Mbps to avoid severe delay. The original BEV feature map is `64 x 200 x 200` in size, corresponding to 9.77 MB. The result-level fusion utilizes the output segmentation and offset maps, which are `2 x 200 x 200` in size, corresponding to 0.305 MB. We calculate the transmission expenses for each method, as shown in Table VI of the attached PDF.
>
> All partially connected methods of intermediate collaboration meet the above bandwidth requirements. Notably, although  V2X-ViT and V2VNet achieve better performance, their high transmission expenses make them less feasible for practical applications. By the way, our proposed DHD has shown the best performance within the partially connected approaches for intermediate collaboration, so we highlight it in bold.
>
> [1] Arena F, Pau G. An overview of vehicular communications[J]. Future internet, 2019, 11(2): 27.
> ### [Weakness 3]: Exploring the number of drones varies from 1 to 4?
> To explore how the number of collaborating drones affects performance, we conduct relevant experiments, as shown in the overall rebuttal.
>
> For the short-range area of 50mx50m, three drones are sufficient to predict trajectories, with performance comparable to that of four drones. However, for the long-range area of 100mx100m, predictive performance improves as the number of drones increases.
> This difference arises because the drones in our dataset are positioned near the intersection to monitor traffic flow, making it easier to cover areas close to the intersection, which fall within the short-range area. In contrast, much of the information in the long-range area extends along specific road branches, which are only partially captured by the drones at the intersection. Therefore, having more drones results in more comprehensive coverage of the long-range area.
> Conversely, the reduction in the number of drones does indeed reflect the influence of potential drone failures on performance.
> ### [Weakness 4]:  Explore how the extent of extrinsics bias affects collaborative prediction?
> We have conducted a series of experiments by introducing Gaussian noise to the extrinsic parameters of drone cameras, as shown in Fig. 2 of the attached PDF document and described in Answer 1 of the overall rebuttal.
> ### [Question 1]:  More details about Air-Co-Pred? Why are totally four drones?
> The dataset is gathered using four drones, each with a field of view set to 70 degrees by default. The drones are positioned at an altitude of 50 meters, monitoring traffic flow at urban intersections. The speed distribution of the observed objects ranges from 0 to 10 m/s, with low-speed states primarily concentrated around 1.5 m/s and high-speed states around 8 m/s. In the CARLA simulations, speed is influenced by several factors, including the default safe speed settings and the necessity of complying with traffic regulations and preventing collisions.
>
> Furthermore, the four drones are arranged in a square formation to achieve a square-shaped coverage area for comprehensive aerial observations. This configuration allows for overlapping fields of view to mitigate occlusions, and the unique observations from each drone enrich the out-of-sight perception capabilities of the entire system.
> ### [Question 2]:  The detailed derivation process.
> According to the pinhole camera model, the transformation between  the world coordinate system and the pixel coordinate system can be described by :
> $$
> \begin{bmatrix}
> 	x \\\\
> 	y \\\\
> 	z
> \end{bmatrix} = {R}^{-1} \\left( {K}^{-1} \begin{bmatrix}
> 	u \\\\
> 	v \\\\
> 	1
> \end{bmatrix} {d} - {T} \\right).
> $$
>
> Next, we consider \\({M} = {R}^{-1}{K}^{-1} \\) and \\( {N} = {R}^{-1}(-{T}) \\)  for further simplification and derivation:
> $$
> \begin{bmatrix}
> 	x \\\\
> 	y \\\\
> 	z
> \end{bmatrix} = {M} \begin{bmatrix}
> 	u \\\\
> 	v \\\\
> 	1
> \end{bmatrix} d + {N} = \begin{bmatrix}
> 	m_{11} & m_{12} & m_{13} \\\\
> 	m_{21} & m_{22} & m_{23} \\\\
> 	m_{31} & m_{32} & m_{33}
> \end{bmatrix} \begin{bmatrix}
> 	u \\\\
> 	v \\\\
> 	1
> \end{bmatrix} d + \begin{bmatrix}
> 	n_1 \\\\
> 	n_2 \\\\
> 	n_3
> \end{bmatrix}.
> $$
> By setting \\( y = -H \\),
> the depth upper-bound can be derived as:
> $$
> {D_{(u,v)}}=\frac{-H-n_2}{m_{21} u+m_{22} v+m_{23}}.
> $$
>
> ### [Question 3]: Clarify the differences with related works.
> Indeed, these two works leverage the camera parameters of UAVs to enhance cross-view association. Specifically, they project 2D detection results from each perspective into a unified 3D space with camera parameters and associate targets based on feature similarity and spatial distance.
> In contrast, our proposed GBG module uses UAV camera parameters along with the available flight altitude to derive the depth upper bound of each pixel.
> This geometric prior helps to constrain depth estimation and calculate the viewing angle of each pixel, facilitating the subsequent BEV generation.
> By acknowledging these works and clarifying our distinct approach, we highlight the unique contributions and applications of the GBG module in multi-drone collaboration.
> ### [Question 4]: Additional details or codes.
> We have already provided the core  code in the initial supplementary materials. Additionally, to ensure better understanding and reproducibility, we plan to release the Air-Co-Pred dataset, the source codes of the complete project, and the model checkpoints, which include our methods and previous baselines. This will contribute to further reproduction and improvement of the system.

---

> > ### Comment · Reviewer_vjyF · 2024-08-09
> > **Comment after reading the response**
> >
> > Thanks for the authors’ time and effort in responding to the weaknesses and questions raised.
> >
> > Overall, this revised version is more comprehensive, incorporating some previously missing details and effectively resolving all my concerns. Moreover, the innovations presented in the paper have been fully recognized by reviewers CzWU, nEkT, and rEnT, and partially acknowledged by reviewer nhVA. Consequently, I am convinced to maintain my initial judgment.

---

> > > ### Author Response · Authors · 2024-08-12
> > > **Thank you for the feedback**
> > >
> > > Thank you very much for taking the time to read our responses and provide your valuable feedback. As mentioned, we are committed to sharing our resources with the community. If the submission is accepted, we will release the dataset, along with the relevant codes and model checkpoints, to facilitate further research and development.

---

### Official Review · Reviewer_rEnT · 2024-07-09

**Soundness:** 3
**Presentation:** 3
**Contribution:** 2
**Rating:** 6
**Confidence:** 2

**Summary:**

This paper studied the problem of collaborative multi-drone object trajectory prediction. The authors proposed a framework named "Drones Help Drones" (DHD) that can improve accuracy compared to existing methods while reducing the required communication bandwidth. To be more specific, the authors leveraged a Ground-prior-based BEV Generation module that can generate more accurate BEV representations, and used a Sparse Interaction via Sliding Windows module that enables efficient information sharing between drones. The authors also constructed a new dataset named "Air-Co-Pred" for multi-drone collaborative prediction tasks.

**Strengths:**

The paper is well-structured and clearly written, with helpful visualizations that aid in understanding the proposed methods and results. The DHD framework introduces a solution to key challenges in multi-drone collaborative perception, particularly the GBG module for more accurate BEV generation and the SISW module for efficient information sharing. The introduction of the "Air-Co-Pred" dataset addresses a gap in existing resources for multi-drone collaborative prediction tasks, potentially benefiting the broader research community.

**Weaknesses:**

* The primary evaluation is conducted on simulated data. While the authors acknowledge this limitation, it raises questions about the framework's performance in more complex, real-world scenarios with sensor noise, communication delays, environmental variability, and other common difficulties in real-world experiments.
* It would be interesting to see how the performance scales with the number of collaborating drones, which could provide valuable insights into the method's scalability. Is there an optimal or maximum number of drones for effective collaboration?

**Questions:**

see above

**Limitations:**

Limitations are discussed

---

> ### Author Rebuttal · Authors · 2024-08-06
>
> ### [Weakness 1]: Questions about the framework's performance in more complex, real-world scenarios with sensor noise, communication delays, environmental variability, and other common difficulties in real-world experiments.
>
> To address the reviewer’s concerns about the gap between real-world scenarios and simulated data, we have incorporated some of the mentioned challenges into our experimental setup to assess their impact on the performance of multi-drone collaborative prediction. Specifically, we examine the effects of sensor noise, which primarily affects the acquisition of camera extrinsics, as well as environmental factors such as drone vibrations and rough terrain, which can influence the drone's relative height to the ground plane. To evaluate the impact of these factors on performance, we conduct complementary experiments by introducing Gaussian noise into these aspects. The statistical results are shown in the attached PDF, and a detailed analysis is provided in Answer 1 of the overall rebuttal.
>
> As to communication delays, we propose integrating a delay assessment module into the collaborative interaction stage of DHD. This module would compensate for lost or delayed frames by leveraging historical collaborative features. Moreover, the compensation process can be streamlined by prioritizing critical areas using the information volume assessment component in the SISW module. At the downstream decoder, missing predictions can also be filled by blending predictions from adjacent time frames and extrapolating trends accordingly.
>
> ### [Weakness 2]: How does the performance scale with the number of collaborative drones? Is there an optimal or maximum number of drones for effective collaboration?
>
> To explore how the number of collaborating drones affects performance, we conducted relevant experiments, as shown in the table below:
>
> | Num of Drones  | IoU (Short) | IoU (Long) | VPQ (Short) | VPQ (Long) |
> |------|-------------|------------|-------------|------------|
> | 1    | 41.5        | 31.1       | 33.5        | 25.6       |
> | 2    | 57.1        | 45.4       | 48.2        | 38.4       |
> | 3    | 61.3        | 50.2       | 50.4        | 43.4       |
> | 4    | 61.4        | 54.0       | 50.4        | 46.2       |
>
> For the short-range area of 50m x 50m, three drones are sufficient to predict trajectories, with performance comparable to that of four drones. However, for the long-range area of 100m x 100m, predictive performance improves as the number of drones increases.
>
> This difference can be attributed to the positional layout of the drones in the dataset, Air-Co-Pred. The drones are situated near intersections to monitor traffic flow, facilitating comprehensive coverage of the short-range areas close to these intersections. In contrast, much of the information in the long-range areas extends along specific road branches, which may only be captured by a single drone. Therefore, increasing the number of drones results in more comprehensive coverage of the long-range areas.

---

> > ### Comment · Reviewer_rEnT · 2024-08-12
> >
> > I would like to thank the authors for their detailed explanation. I've raised the score accordingly.

---

> > > ### Author Response · Authors · 2024-08-12
> > > **Thank you for the feedback**
> > >
> > > Thank you for patiently pointing out the weaknesses in our initial manuscript and reviewing our response. If the submission is accepted, we will include a detailed analysis of our proposed DHD's real-world limitations and scalability in the revised version, ensuring a more comprehensive presentation of our paper.

---

### Official Review · Reviewer_nhVA · 2024-07-12

**Soundness:** 2
**Presentation:** 3
**Contribution:** 2
**Rating:** 5
**Confidence:** 3

**Summary:**

The paper presents the "Drones Help Drones" (DHD) framework for improving collaborative trajectory prediction in multi-drone systems. It addresses challenges with aerial observations and communication bandwidth by: (i) Using ground priors from inclined drone observations to enhance Bird's Eye View (BEV) accuracy; (ii) Implementing a selective mechanism to prioritize essential information, reducing communication needs; (iii) Introducing the "Air-Co-Pred" dataset for multi-drone collaborative prediction.
Experiments show that DHD improves BEV accuracy, reduces required transmission bandwidth, and maintains high prediction performance.

**Strengths:**

The paper presents an innovative approach to multi-drone collaborative trajectory prediction. Here are some observations:

S1. The module for estimating depth from multi-view drone perspectives is convincing and well-formulated. The use of object height contributes significantly to research in this field.

S2. The strategy requires only 25% of the transmission ratio without affecting performance. This might be due to the SISW module, which retains only essential information.

S3. The three strategies (depth estimation, interaction, and prediction) are trained end-to-end. This is somewhat surprising given the architecture design.

**Weaknesses:**

W1. The strategy follows the classic collaborative framework composed of three modules: BEV estimation, construction of interaction, and prediction. The methodology utilizes well-known strategies from the literature for this task. Specifically, for BEV estimation, it uses part of the framework from [13], although it is well adapted with the inclusion of height in the depth estimation. The Selective Sparse Interaction used in the SISW module appears similar to the approach in [14], as well as the aggregation performed. Finally, the prediction mechanism is taken from [34], as mentioned by the authors in L206.
Although the construction of a unified framework is not trivial, the overall strategy deviates little from existing ones.

W2. The authors have contributed to the creation of a new dataset that could benefit future research, but this will require appropriate adjustments. There is no comparison with other datasets used in the same or related fields, making it impossible to evaluate the quality of the dataset.

W3. The lack of additional datasets also makes the effectiveness of the framework less convincing, as there is no evidence of generalization on other established benchmarks. I understand the uniqueness of the task and wonder if it is possible to test the framework on established benchmarks (e.g., DAIR-V2X, V2XSet, and OPV2V datasets).

W4. In Table 2, the reasoning for separating V2X-ViT and V2VNet from other methodologies is not clear. The distinction between fully/partially connected is not a convincing rationale. Could the authors clarify this point?

W5. [Minor] Some findings in the appendix, such as Figure 9 and Section 6.4, could have been included in the main paper to strengthen certain claims.

**Questions:**

See Weaknesses section for questions, especially I would appreciate if the authors highlighted the differences with the other mentioned methodologies.

**Limitations:**

The authors adequately addressed the limitations.

---

> ### Author Rebuttal · Authors · 2024-08-06
>
> ### [W1 & Q1]: Differences with existing methodologies.
> Our DHD builds on existing solutions, but we make significant improvements to address unique challenges in multi-drone collaborative prediction.
> For instance, long-range aerial observations can lead to substantial depth estimation errors, affecting BEV representations. Additionally, objects in view are typically small and sparse, with critical information making up only a minor portion. However, previous methods, primarily focusing on collaborative detection, have not adequately considered the selection and utilization of sparse information for joint prediction. Our DHD framework is specifically designed to overcome these challenges, distinguishing our approach from others.
>
> Regarding the drone-specific BEV generation module, GBG, while it is based on [13], we introduce innovations of geometric priors to refine depth estimation, as acknowledged by Reviewer CzWU rEnT vjyF. Moreover, the improvements we've made in the LSS method are also applicable to other depth-based BEV generation techniques.
> As for the similarity with [14], our SISW module focuses on areas with significant feature variations for subsequent collaborative feature fusion, which are likely to contain objects or critical environmental information essential for prediction tasks. This solution is not considered by previous collaborative strategies, including [14]. In terms of implementation, the sliding window mechanism for information volume assessment is a major contribution of our SISW module, as noted by Reviewer CzWU nEkT. Additionally, for the selective sparse interaction component, we optimize the selection mechanism by replacing traditional threshold-based filtering with score ranking. This modification allows us to precisely control transmission volume based on predefined ratios, rather than relying on manually-tuned empirical thresholds.
>
> For the feature fusion module, while not our primary innovation, our approach differs significantly from [14].  Instead of their attention-based fusion, we use a multi-layer convolutional method to calculate contribution weights for each collaborative feature guided by local features. Theoretically, attention mechanisms are better suited for global feature associations, while the convolutional paradigm is widely recognized for its strength in preserving local features. Since our instance-level prediction task benefits more from strong local feature representations, we opt for this fusion method and provide additional experiments to demonstrate performance differences.
>
> ### [Weakness2]: A comparison with other datasets.
> Please refer to Answer 3 of the overall rebuttal. Additionally, we provide a detailed comparison of key attributes in the table below:
>
> | **Attribute**                | **VisDrone-MDOT** | **VisDrone-MDMT** | **Airsim-Map** | **CoPerception-UAVs** | **Air-Co-Pred**       |
> |------------------------------|-------------------|-------------------|---------------|-----------------------|-----------------------|
> | Source                       | TCSVT-21          | TMM-23            | CVPR-20       | Neurips-22            | Submit to Neurips-24  |
> | Real/Simulation              | Real              | Real              | Simulation    | Simulation            | Simulation            |
> | Support Tasks                | 2D SOT            | 2D MOT            | 2D Seg        | 2D/3D Det, BEV Seg    | 2D/3D Det, BEV Seg, Pred, MOT |
> | Number of Drones             | 2~3               | 2                 | 5             | 5                     | 4                     |
> | Number of Samples            | 105,584           | 19,839            | 4,000         | 5,276                 | 8,000                 |
> | Frequency                    | 10Hz              | 10Hz              | Unknown       | 0.25Hz                | 2Hz                   |
> | Cam. Params. & Coord. Info.  | x  | x  | x  | √ | √  |
> | Suitability & Justification  | x; w/o Cam. Params. & Coord. Info. | x; w/o Cam. Params. & Coord. Info. | x; w/o Cam. Params. & Coord. Info. & Temporal Annotations.| Partially; Long time intervals render it unsuitable for temporal tasks. | √ |
>
> Cam. Params. & Coord. Info. refers to camera parameters and coordinate information, which are critical for BEV generation.
> ### [W3]: Additional datasets. (e.g., DAIR-V2X, V2XSet and OPV2V datasets).
> The benchmarks you mentioned are primarily designed for collaboration in autonomous driving scenarios, where the viewing angles of vehicles are horizontal. This contrasts sharply with the aerial oblique perspectives in our multi-drone collaboration research, making our drone-specific BEV generation module, GBG, less applicable to these benchmarks.
> Besides, in the initial manuscript, we provided experimental results on the publicly available CoPerception-UAVs dataset in Table 4 of the initial manuscript, to demonstrate the generalization capabilities of our framework for collaborative detection.
>
> We understand your concern about the reliance on simulated multi-drone datasets. In fact, we are currently developing a real-world multi-drone collaboration dataset to further validate and extend our work in future studies.
> ### [W4]: Reasons for separating V2X-ViT and V2VNet from others.
> Benefiting from the fully connected paradigm, V2X-ViT and V2VNet achieve complete feature-level interaction among collaborators and employ more complex techniques for collaborative feature fusion. As a result, it is reasonable for them to demonstrate superior performance compared to the partially connected paradigm.
>
> From the perspective of communication overhead, it is logical to distinguish V2X-ViT and V2VNet from other communication-efficient methods for a fair comparison.
> Notably, our proposed SISW outperforms current partially connected methods and is comparable to the fully connected paradigm.
> ### [W5]:
> Thank you for your suggestion regarding the content layout.

---

> > ### Comment · Reviewer_nhVA · 2024-08-12
> >
> > Thank you to the authors for the thorough explanation of the differences with existing methodologies. I also appreciated the comparison of the dataset with others in the literature; it would be helpful if this could be included in the revised version of the paper.
> >
> > In general, my concerns have been addressed, and I am inclined to raise the score.

---

> > > ### Author Response · Authors · 2024-08-12
> > > **Thank you for the feedback**
> > >
> > > Thank you very much for reading our responses! If the submission is accepted, we will incorporate more details about the differences between our DHD and existing methodologies, and highlight the comparison of our proposed Air-Co-Pred with other datasets in the final version.

---

### Official Review · Reviewer_nEkT · 2024-07-14

**Soundness:** 2
**Presentation:** 3
**Contribution:** 2
**Rating:** 4
**Confidence:** 4

**Summary:**

The paper proposes a ground-prior-based BEV generation for drone trajectory prediction. The methods also propose an efficient information interaction via a Sliding Windows module. It accesses information volume across different areas through sliding windows. The authors provide experiments on the simulator-based Air-Co-Pred dataset and 3D object detection on CoPerception-UAVs.

**Strengths:**

1.	The flow of the paper is thorough and clear.
2.	The proposed ground-prior-based BEV generation and Sliding Windows module are innovative.

**Weaknesses:**

The Ground-prior-based BEV Generation (GBG) module relies on certain geometric assumptions for depth estimation. In real-world scenarios, these assumptions might not hold due to variations in terrain, drone stability, and other environmental factors, potentially affecting the accuracy of BEV representations.


Although the paper mentions using EfficientNet-B4 for feature extraction due to its low latency and optimized memory usage, it does not provide a detailed analysis of the computational efficiency and scalability of the overall framework.


The Air-Co-Pred dataset, while a valuable contribution, is still limited to simulated data. The paper could benefit from additional datasets or validation for the main task.


The proposed Sparse Interaction via Sliding Windows (SISW) module, although innovative, might oversimplify the information selection process. The method assumes that feature discrepancies are the primary indicators of valuable information, which may not always be the case, especially in highly dynamic or cluttered environments.


There is a lack of baseline in the main experiment; the authors have compared their method with only two other methods for BEV generation.

There is a lack of trajectory prediction evaluation; the authors should provide the average displacement error as well.

Typo in Table 3 caption.

SISW has shown little improvement, and the performance declines when the transition rate is 0.25. Additionally, the authors do not provide the individual result of SISW on 1 with GBG still present.

**Questions:**

Can the authors provide a component ablation of only the SISW component, like three rows on a 1:1 ratio? Furthermore, the results of missing GBG and SISW on a 0.25 transition rate are not provided.

**Limitations:**

The paper relies on simulations using the CARLA platform. Although the simulated environment provides controlled and varied scenarios, it may not fully capture the complexities and unpredictabilities of real-world environments.

It seems like the sliding window module is not that significant, and the authors have not provided a complete ablation study to show its efficacy.

The component ablation is not thorough.

---

> ### Author Rebuttal · Authors · 2024-08-06
>
> ### [W1 & L1]: Consider variations in terrain, drone stability, and other environmental factors.
> Please refer to Answer 1 in the overall rebuttal.
> ### [W2]: Analysis of the computational efficiency and scalability of the overall framework
> As seen in Table VI of the PDF, the differences in trainable parameters (which reflect model complexity) and model parameter sizes (which impact memory usage) among the compared methods are not significant, except for Who2com and When2com. Notably, the FLOPs for V2VNet and Late Collaboration are significantly higher than those of other baselines. This is due to the iterative processes involving multiple for-loops for information exchange, which generate multiple parallel paths during computation. Each branch performs independent calculations before merging the results, significantly increasing computational complexity. The same principle applies to the difference in FLOPs between our DHD framework and Where2comm.
>
> For the investigation of stability, we explore the number of collaborative drones in Answer 2 of the overall rebuttal.
> ### [W3]: Additional datasets for validation.
> Please refer to Answer 3 in the overall rebuttal.
> ### [W4]: SISW might oversimplify the information selection process. Can it handle highly dynamic or cluttered environments?
> Our proposed SISW module is specifically designed with the characteristics of aerial perspectives, where targets are often small and sparse, and surrounding environmental textures can guide trend prediction. Specifically, SISW employs a sliding window mechanism to assess and select critical information. This design is simple yet efficient, outperforming previous sparse interaction baselines.
>
> Although the SISW module is not primarily designed for highly dynamic or cluttered environments, our dataset, Air-Co-Pred, inherently includes such scenarios, thereby enhancing the module's adaptability and robustness.
> Specifically, our dataset contains substantial moving vehicles, which aligns with the highly dynamic scenarios mentioned.
> As illustrated in Figure 6 of the initial manuscript, the low proportion of targets within the field of view indicates a high presence of clutter.
> Furthermore, our adopted BEV representations help filter out much of the clutter in 2D vision during spatial projection by truncating the z-axis.
>
> Therefore, the SISW module proposed in this paper can be considered applicable to dynamic and cluttered environments to a certain extent.
> ### [W5]: A lack of BEV baselines.
> Thank you for your feedback. In our work, we focus on optimizing depth estimation from aerial perspectives within the BEV generation process. The improvements we've made in the GBG module are indeed applicable to other depth-based BEV generation methods.
> We recognize the importance of comparing our method against a broader range of baselines.
>
> To provide additional comparisons, we include methods like **BEVDet4D**, which considers temporal BEV representations, and **BEVerse**, which applies multi-task learning to improve BEV generation. As shown in Table V of the PDF, while multi-task learning offers minimal performance gains, incorporating temporal information does enhance BEV generation. However, our DHD framework with the GBG module, which leverages geometric priors, still outperforms these methods.
> ### [W6]: Calculate the average displacement.
> As illustrated in Table II of the PDF, a lower ADE does not necessarily indicate better prediction performance. For example, "No Collaboration", which relies on single-view observations, misses a significant number of objects. However, its calculated ADE is relatively low because this metric only considers matched trajectories. Similarly, methods that performed well in our initial experiments, such as Early Collaboration, V2VNet, V2XViT, and our proposed DHD, exhibit higher ADE compared to Where2comm and UMC. This is partly due to Where2comm and UMC having a higher missing ratio, which excludes some difficult trajectories from the ADE calculation. Therefore, when using ADE as a metric, it's important to also consider the ratio of missing trajectories.
> This is why we adopt evaluation metrics such as IoU and VPQ, as suggested in the research of our adopted predictor ''PowerBEV''. These metrics provide a more comprehensive assessment by considering missed detections, false positives, and the deviation between predicted and ground truth positions.
>
> ### [W7]: Typo errors.
> We have noted the incorrect usage of quotation marks and will correct it in the revised version.
> ### [W8 & Q1 & L2 & L3]: Additional component ablation study about SISW module.
> Regarding your comment that "SISW has shown little improvement, and the performance declines when the transition rate is 0.25," the SISW module is specifically designed to preserve critical information for the prediction task during sparse collaborative interactions. This concept is also positively acknowledged in Strength 2 of Reviewer nhVA's feedback. Given that our method transmits only a subset of features, it is understandable that there may be a slight performance decline compared to the complete feature transmission.
>
> We have enriched the ablation studies as you suggested in Table IV of the PDF. Notably, when the SISW module is absent, we resort to randomly selecting 25% of spatial features for transmission to achieve spatial sparsity.
> The additional experiments show that when the transmission ratio is close to 1, our DHD framework, which includes both the GBG and SISW modules, provides a performance gain compared to the variant with only GBG, particularly in the VPQ metric. This improvement is attributed to the collaborative feature aggregation component in SISW, which enhances the weight of features that are beneficial for downstream prediction tasks.
>
> The enhanced ablation experiments, which consider both module-level and transmission-level factors, clearly demonstrate the effectiveness of both the GBG and SISW modules.

---

> ### Author Response · Authors · 2024-08-14
>
> Dear Reviewer nEkT. We hope that our rebuttals have clarified your concerns. if there are any specific analyses or complementary experiment that could clear your doubts, we would be happy to try and provide them. We sincerely thank you again for your time and feedback.

---

### Official Review · Reviewer_CzWU · 2024-07-25

**Soundness:** 3
**Presentation:** 3
**Contribution:** 3
**Rating:** 5
**Confidence:** 4

**Summary:**

This paper proposes a framework, Drones Help Drones (DHD), which tackles trajectory prediction of objects in the scene. DHD consists of a Ground Prior Based Bird’s Eye View (BEV) Generation (GBG) module, which provides depth estimation from the drone to an object using ground priors to create an accurate BEV representation of the features. It also utilises Sparse Interaction via Sliding Window (SISW) to minimize the data transmission cost between drones. In addition, the authors develop a new dataset for multi-drone collaboration prediction, "Air-Co-Pred". The paper is interesting in the field of collaborative AI.

**Strengths:**

The paper is well-structured, referencing a well-established research stream in computer vision and collaborative object trajectory prediction. The following are some strengths of the paper:
- The research problem is clearly described with good visuals and diagrams, which aid in the explanation.
- There are several contributions, including BEV generation, sliding windows for sparse interaction, and Air-Co-Pred simulated dataset
- The paper provides both quantitative and qualitative assessments of their framework, compared to baseline and state-of-the-art, such as Who2com and Where2com. It demonstrates good improvements.
- The provided appendices are useful for further details and ablation studies.

**Weaknesses:**

The paper has several weaknesses:
- It investigates methods to overcome single-drone issues, such as occlusions and blurs; nevertheless, it is also important to discuss a bigger picture of their use cases, including accident prevention and path planning, in greater details with certain limitations. For example, in accident prevention, if multiple drones collaborate and predict an accident is about to happen, what can it do? Does it then communicate/inject commands over the air to the vehicle causing certain actions?
- The DHD framework consists of feature extraction, BEV, Sparse Interaction via Sliding Windows, and Trajectory Prediction. The idea of having sliding windows for sparse interaction is quite interesting; nevertheless, relying on a local coordinate system and pixel-level weight fusion can be a weak spot when it comes to real-world settings. It is also important to examine how each of the modules contributes to the overall performance in ablation studies.
- The development of Air-Co-Pred remains questionable, based on CARLA. There are indeed many CARLA-based datasets and it is important to compare your dataset with others to prevent overlappings/duplications.
- It is vital to discuss the following papers, in relation to the work:
1. Wei, S., Wei, Y., Hu, Y., Lu, Y., Zhong, Y., Chen, S., & Zhang, Y. (2024). Asynchrony-robust collaborative perception via bird's eye view flow. Advances in Neural Information Processing Systems, 36.
2. Lu, Y., Hu, Y., Zhong, Y., Wang, D., Chen, S., & Wang, Y. (2024). An extensible framework for open heterogeneous collaborative perception. arXiv preprint arXiv:2401.13964.
3. Liang, J., Jiang, L., & Hauptmann, A. (2020). Simaug: Learning robust representations from simulation for trajectory prediction. In Computer Vision–ECCV 2020: 16th European Conference, Glasgow, UK, August 23–28, 2020, Proceedings, Part XIII 16 (pp. 275-292). Springer International Publishing.

**Questions:**

There are many questions to address as the following:
- With the DHD framework, given objects to track/predict, what is the optimal number of drones that need to be “watching” the object in order to produce the best prediction? Is it always the case that the more drones that have the object in view, the better the trajectory prediction? Why four collaborative drones in Air-Co-Pred? Can we get away with just 2 drones?
- Can you explain more about why CARLA is used to simulate and produce the dataset? Has anyone attempted trajectory prediction with multiple drones in real-world settings? Is it feasible? What needs to happen before DHD can be deployed in production in the real-world setting? Maybe it can be included in a discussion section.
Is there an optimal drone height/altitude? Is 50 meters the best height value, so it’s used in the dataset?
- How often should the drones be communicating/transmitting data to each other? Is it the same as the aerial observation samples being collected (frequency of 2 Hz)? Can it be reduced to further lower transmission data?
- How does the DHD framework respond to noises, such as flight turbulence?
- What will be the performance if we turn on/off multiple components, such as BEV or Sparse Interaction? This should be included in ablation studies.
- How far are we from real-world experiments?
- What will be the impacts of your study to the field?

**Limitations:**

The authors have raised concerns about the use of simulated settings, which limit the research's practicality. It is important to develop real-world scenarios for future validations of the research.

---

> ### Author Rebuttal · Authors · 2024-08-06
>
> ### [W1]:
> We envision a scenario where vehicles are centrally coordinated. In this setting, collaborative drones predict abnormal trajectories that may lead to accidents. Upon identifying the risks, the drones use wireless communication to alert nearby vehicles, prompting them to take evasive actions to prevent accidents. The system can also integrate with LLMs to provide solutions. Besides, it can be used to forecast traffic congestion, offering lane recommendations.
> ### [W2]:
> As to the concern about obtaining UAV coordinate systems in real-world scenarios, these parameters are considered readily available data sources in [1].
>
> Regarding the pixel-level weight fusion in SISW, the reviewer's concern may stem from the computational overhead. Our approach focuses on fine-grained feature fusion, which is indeed more complex than simple addition or concatenation. As shown in TABLE III of the PDF, we supplement the results with common fusion methods. As expected, they perform worse than ours.
>
> [1] Zhao M, Chen J, Song S, et al. Proposition of UAV multi-angle nap-of-the-object image acquisition framework based on a quality evaluation system for a 3D real scene model of a high-steep rock slope[J].
> ### [W3]:
> Below is a description of other CARLA-based datasets.
>
> **For Autonomous Driving:**
> - **OPV2V**: Focuses V2V collaborative perception to enhance safety and efficiency in autonomous driving.
> - **OPV2V-Occupancy**: An extension of OPV2V, it introduces an occupancy grid map to represent the distribution of surrounding obstacles.
> - **V2X-Sim**: Simulates both V2V and V2I modes, covering a range of complex scenarios. However, it does not simulate noise environments and includes only a single type of road.
> - **V2XSet**: Simulates real-world noise in V2X collaboration.
> - **V2XSet-w**: An extended version of V2XSet, it includes more adverse weather conditions.
> - **DeepAccident**: Generates various traffic accident scenes and records relevant sensor data. This dataset is intended for traffic accident analysis and prevention.
>
> **For UAV Research:**
> - **CoPerception-UAVs**: Refer to Answer 3 in the overall rebuttal.
>
> It is evident that the key difference between our Air-Co-Pred and other CARLA-based datasets is our aim to solve fundamentally different problems from the outset. Furthermore, in terms of dataset characteristics, unlike the horizontal perspective of autonomous driving scenes, our multi-drone dataset focuses on aerial observations with oblique views over wide ranges. Additionally, compared to CoPerceptionUAVs, our dataset supports additional temporal tasks such as prediction and tracking.
> ### [W4]:
> - **CoBEVFlow**: It addresses the challenge of asynchrony in collaborative perception by focusing on robustness to temporal misalignments. Although our current work does not account for asynchrony, the proposed solutions in their study provide valuable insights.
> - **HEAL**: Lu et al. propose a framework for heterogeneous collaborative perception, focusing on collaboration among diverse data sources and perception models. Our work could benefit from their approach when dealing with heterogeneous members in collaboration.
> - **SimAug**: This research explores learning robust representations for trajectory prediction using rich simulated data from CARLA. Specifically, they identify hard examples from multi-view information and combine them with the original view through adversarial learning to better adapt to real-world environments.
> ### [Q1]:
> Refer to Answer 2 of the overall rebuttal.
> ### [Q2]:
> - **Why CARLA?**
>   CARLA offers the flexibility to create complex urban traffic scenarios, providing rich dynamic objects along with automated annotations, including the camera intrinsics, extrinsics, and coordinate information needed for BEV generation. It can generate extensive data for algorithm validation with minimal time and manpower investment.
> - **Is it feasible for multi-drone trajectory prediction?**
>   To the best of our knowledge, real-world trajectory prediction with multiple drones is still in an early stage. The most relevant multi-drone collaborative datasets, such as VisDrone-MDOT and VisDrone-MDMT, are primarily collected for tracking. However, their temporal continuity and ID information make them adaptable for trajectory prediction, indicating the feasibility of using multiple drones for joint prediction.
> - **Why 50m?**
> At this altitude, four drones near intersections can collaboratively cover an area of around 100mx100m. This height also results in significant occlusions, such as vehicles being obscured by buildings or trees, which we aim to address through multi-drone collaboration.
> ### [Q3]:
> We set the interaction frequency to match the sampling frequency of 2 Hz. However, we can explore reducing this by leveraging recent multi-drone collaboration predictions, such as instance segmentation and offsets, to extrapolate trends and strategically fuse them with current single-view predictions.
> ### [Q4]:
> Refer to Answer 1 of the overall rebuttal.
> ### [W2.1 & Q5]:
> Refer to TABLE IV of the PDF.
> ### [Q6 & Q2.4]:
> Before DHD can be deployed in real-world applications, several steps are necessary:
> - **Robustness**: Can handle sensor noise and communication delays, which are common in real-world scenarios.
> - **Scalability and Flexibility**: Adapt to the varying number of drones and remain resilient to drone failures or disconnections.
> - **Environmental Adaptability:**: Account for weather conditions, lighting variations, and complex terrains.
> - **Real-Time Performance**: Optimize for lightweight deployment on edge devices with high FPS.
> ### [Q7]:
> To the best of our knowledge, we are the first to employ an end-to-end approach that combines multi-drone joint observations for trajectory prediction. Additionally, we provide a simulated multi-drone dataset to support this work, which can be used for various tasks and serves as a benchmark for future research in multi-drone collaboration.

---

### Author Rebuttal · Authors · 2024-08-06

## Answer 1: Investigate the effects of sensor noise, flight turbulence and rough terrain on the performance of DHD.

### Flight Vibrations and Uneven Terrain.
We acknowledge that flight vibrations and uneven terrain can interfere with the drone's relative height to the ground, affecting the BEV generation from the GBG module. Therefore, we introduce perturbations to the drone's altitude to simulate these conditions.

Specifically, we introduce Gaussian noise to the drone's altitude, with noise levels ranging from 0.002 to 0.01. At the highest level, this results in a maximum altitude variation of 0.5 meters, which is significant for drone flight. As illustrated in Fig. 1 of the attached PDF, the results demonstrate that when the noise level exceeds 0.01, the depth estimation advantage conferred by the geometric prior in the GBG module diminishes.
However, this limitation of terrain variations is not unique to our GBG module. Mainstream BEV generation methods, such as LSS, BEVFormer, and PETR, also assume a flat ground. Complex terrain requires further study and can be considered a distinct research direction.

The primary goal of our GBG is to explore BEV generation specifically designed for drones. While this is an initial attempt, we recognize the need to account for more complex real-world conditions in future deployments. To address this, we propose a simple yet potentially effective solution: developing a ground flatness estimation module to assess variations in the ground plane, allowing the estimated object height to be adaptively adjusted and thereby mitigating the impact of uneven terrain on subsequent BEV generation.

### Sensor Noise, Particularly in Extrinsic Parameters.

As illustrated in Fig. 2 of the PDF, increasing noise results in a gradual decline in both IoU and VPQ. In short-range settings, noise ratios below 0.003 cause negligible performance drops. However, IoU decreases by about 25% when the noise ratio is between 0.003 and 0.013. Between 0.013 and 0.020, the decline slows, with an additional reduction of approximately 10%. VPQ exhibits a similar trend.

In long-range settings, noise ratios below 0.003 also result in acceptable performance declines. However, when the ratio reaches 0.005, noticeable performance degradation occurs, with IoU dropping by 21.3% and VPQ by 32.9%. Overall, noise has a more pronounced impact on VPQ than on IoU, indicating that camera extrinsic bias more severely affects the consistency of future trajectory predictions.
Furthermore, the greater impact of extrinsic noise in long-range observations can be attributed to objects at long distances often being observed from a single perspective, lacking the multi-view validation available in short-range scenarios.

These results demonstrate that our DHD can tolerate a small amount of sensor-based extrinsic noise. Besides, larger biases in extrinsic parameters can significantly impact collaborative prediction. Therefore, accurate estimation of these parameters is crucial for maintaining high performance in collaborative perception systems. This finding is equally applicable to real-world scenarios.

## Answer 2: Investigate the number of collaborative drones.
To explore how the number of collaborating drones affects performance, we conduct relevant experiments, as shown in TABLE I of the PDF.

For the short-range, three drones are sufficient to predict trajectories, with performance comparable to that of four drones. However, for the long-range, predictive performance improves as the number of drones increases.

This difference arises because the drones in our dataset are positioned near the intersection to monitor traffic flow, making it easier to cover areas close to the intersection, which fall within the short-range area. In contrast, much of the information in the long-range area extends along specific road branches, which are only partially captured by the drones at the intersection. Therefore, having more drones results in more comprehensive coverage of the long-range area.

## Answer 3: Lack of comparison with other existing datasets.
To demonstrate the contributions of our dataset, Air-Co-pred, we conduct a comparative analysis of existing datasets in the multi-drone collaboration domain. By the submission deadline, several datasets are available for multi-drone collaboration, including two real-world datasets (VisDrone-MDOT and VisDrone-MDMT) and two simulation datasets (Airsim-Map and CoPerception-UAVs).

Regarding the existing real-world datasets, the VisDrone series has collected a substantial amount of real-world video data. However, these datasets are constructed solely from a visual perspective and do not provide any information about the drones' poses or camera parameters. As a result, they are limited to supporting 2D visual algorithms such as ReID and object tracking, and cannot be used to evaluate our proposed collaborative prediction framework, which integrates both visual and spatial information.

The simulation dataset, Airsim-Map, is designed to demonstrate the effectiveness of who2com and when2com in mitigating image degradation. However, it only provides multi-view 2D semantic segmentation masks.
The dataset most similar to ours is CoPerception-UAVs, proposed by where2comm. While this dataset focuses exclusively on multi-drone collaborative 3D object detection and has been used to validate frameworks like Where2comm and CoCa3D, it falls short in addressing joint temporal tasks. Additionally, its large sampling intervals are inadequate for validating our DHD.
To bridge this gap, we propose a more comprehensive dataset that includes detailed annotations and an appropriate sampling frequency. This dataset supports a wide range of tasks, including 2D/3D detection, BEV segmentation, multi-object tracking, and trajectory prediction, and facilitates the preliminary validation of multi-drone collaboration across various scenarios within a simulation environment.

---

### Decision · Program_Chairs · 2024-09-25

**Decision:**

Accept (poster)

**Comment:**

The paper has undergone thorough evaluation by the reviewers, most of whom have raised their scores after the authors addressed their initial concerns. The reviewers are now generally satisfied with the paper, acknowledging its strengths and contributions to the field. After considering the updated reviews and the paper's overall quality, we recommend accepting this submission.